# Electrical and Ca$^{2+}$ signaling in dendritic spines of substantia nigra dopaminergic neurons

**Travis A Hage, Yujie Sun, Zayd M Khaliq\***

Cellular Neurophysiology Unit, National Institute of Neurological Disorders and Stroke, National Institutes of Health, Bethesda, United States

**Abstract** Little is known about the density and function of dendritic spines on midbrain dopamine neurons, or the relative contribution of spine and shaft synapses to excitability. Using Ca$^{2+}$ imaging, glutamate uncaging, fluorescence recovery after photobleaching and transgenic mice expressing labeled PSD-95, we comparatively analyzed electrical and Ca$^{2+}$ signaling in spines and shaft synapses of dopamine neurons. Dendritic spines were present on dopaminergic neurons at low densities in live and fixed tissue. Uncaging-evoked potential amplitudes correlated inversely with spine length but positively with the presence of PSD-95. Spine Ca$^{2+}$ signals were less sensitive to hyperpolarization than shaft synapses, suggesting amplification of spine head voltages. Lastly, activating spines during pacemaking, we observed an unexpected enhancement of spine Ca$^{2+}$ midway throughout the spike cycle, likely involving recruitment of NMDA receptors and voltage-gated conductances. These results demonstrate functionality of spines in dopamine neurons and reveal a novel modulation of spine Ca$^{2+}$ signaling during pacemaking.

**\*For correspondence:** zayd. khaliq@nih.gov

**Competing interests:** The authors declare that no competing interests exist.

## Introduction

The function of dendritic spines, sites of synaptic input for neurons in the central nervous system, is linked intimately to their unique geometry. For example, the thin spine neck limits diffusion of signaling molecules from the spine head (*Svoboda et al., 1996*; *Tonnesen et al., 2014*), a feature critical for synapse specificity during plasticity (*Harvey and Svoboda, 2007*; *Matsuzaki et al., 2004*; *Yuste and Denk, 1995*). Furthermore, theoretical studies hypothesize that the resistance of the spine neck, if high relative to the impedance of the dendrite, may electrically compartmentalize synaptic potentials in the spine head (*Jack et al., 1975*). As a result, the high neck resistance of spines can lead to passive amplification the voltage in the spine head, but also attenuation of signals as they travel across the spine neck (*Araya, 2014*; *Sala and Segal, 2014*; *Yuste, 2013*). Accordingly, long-necked spines may be expected to have higher neck resistances, producing strong attenuation of synaptic potentials. However, experimental evidence for this hypothesis has been less conclusive. Studies of cortical pyramidal cells show a clear negative correlation between neck length and EPSP amplitude (*Araya et al., 2006*; *2014*), while studies in hippocampal pyramidal cells (*Takasaki and Sabatini, 2014*) and olfactory granule neurons (*Bywalez et al., 2015*) observe either a weak relationship or none at all. Therefore, better knowledge of the relationship between spine geometry and synaptic function will forward our understanding of how synaptic input shapes neuronal excitability.

Dopaminergic neurons of the substantia nigra (SNc) are commonly categorized as aspiny neurons, however early anatomical studies are less definitive regarding this view. Spine-like appendages have been reported on dopamine neurons in a variety of species including humans (*Grace and Onn, 1989*; *Kline and Felten, 1985*; *Phelps et al., 1983*; *Preston et al., 1981*; *Rinvik and Grofova, 1970*; *Sarti et al., 2007*; *Schwyn and Fox, 1974*; *Yung et al., 1991*; *Cruz-Sanchez et al., 1995*;

**eLife digest** When a nerve cell is viewed under the microscope, its structure looks a little like that of a tree. Each nerve cell, or neuron, has an array of 'branches' known as dendrites, which receive chemical messages from other cells. These messages are converted into electrical signals in the cell body and then travel down the main nerve fiber, which is the output of the cell. At the end of the nerve fiber, the signals are converted into chemical messages again and passed on to the dendrites of the next neuron.

The dendrites of most neurons are covered with spines that resemble thorns on a stem. These are the sites that typically connect neurons with other cells. However, neurons that release a chemical messenger called dopamine have largely smooth dendrites. These neurons still belong to extensive neural circuits that are involved in movement and reward processing. As such, it is not clear whether dopamine neurons receive connections that form onto dendritic spines, or whether all of their connections are formed directly onto dendrites.

By studying slices of mouse brain, Hage et al. now show that inputs onto dopamine neurons in fact establish both types of connections. The connections formed directly onto dendrites usually have more influence on the neuron than those on spines. However, the size and shape of spines determines their properties. Most spines are attached to dendrites via a narrow neck, and spines with long necks have less influence on the electrical signaling of the cell than short-necked spines. This is because the neck limits the movement of electrical charges. Long-necked spines also possess fewer receptors for the chemical messenger glutamate, which reduces their ability to transmit signals arriving from other cells.

Dopamine neurons receive input from many different areas of the brain. A key next step is to determine whether neurons from specific brain regions are more likely to form connections with spines as opposed to directly on dendrites. Given that these inputs often arrive at the same time, another question is whether there is crosstalk between these two types of connections.

*Patt et al., 1991*). In other studies, spines were encountered only rarely (*Juraska et al., 1977*; *Tepper et al., 1987*). While many of these studies provide important descriptions of spine density, quantitative analyses of the density of spines in SNc dopamine neurons have seldom been performed. Likewise, the function of spines on dopamine neurons is unclear. For example, high densities of vesicular monoamine transporters, dopamine transporters and dopamine D2-autoreceptors have been localized to spines (*Gantz et al., 2015*; *Nirenberg et al., 1996a*; *1996b*), which are associated with the reuptake and storage of dopamine. However, these observations raise the question of whether spines on dopamine neurons also function as typical sites of excitatory synaptic input.

Using two-photon laser scanning microscopy, we examined the density and morphology of dendritic spines on dopamine neurons in live and fixed tissue preparations from juvenile (P6–P25) and adult mice (up to 7 months old). During preparation of this study, a separate study was published that reports a mixture of spine and shaft synapses on dopamine neurons and tests glutamate receptors on spines (*Jang et al., 2015*). Here, we perform a comprehensive analysis of the functionality, chemical and electrical compartmentalization as well as $Ca^{2+}$ signaling in short spines (<2 μm), long spines (2–5 μm) and shaft synapses. In addition, we tested the influence of subthreshold voltage changes on spine $Ca^{2+}$ during slow pacemaking, a characteristic firing pattern of dopamine neurons. We demonstrate the presence of functional dendritic spines on SNc dopaminergic neurons and provide evidence that activation of spines during pacemaking leads to novel enhancement of spine $Ca^{2+}$ that occurs periodically in a window starting at the middle phase and lasts throughout remainder of the spike cycle.

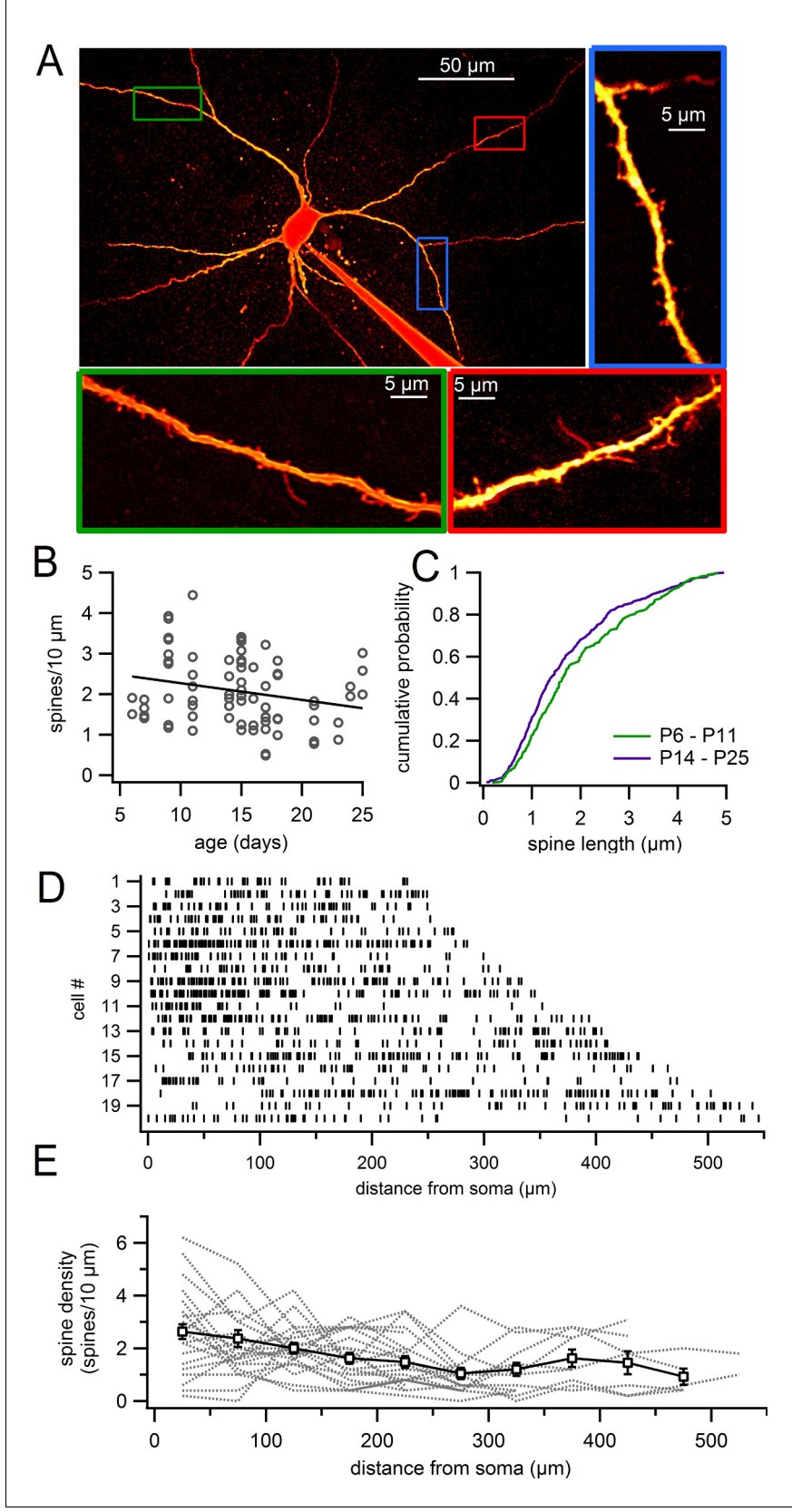

**Figure 1.** Dendritic spines on SNc dopamine neurons visualized in live slices. (**A**) SNc dopamine neuron filled with Alexa-594 via patch pipette, visualized on a two-photon microscope. Higher magnification of selected dendritic

*Figure 1 continued on next page*

*Figure 1 continued*

segments are shown in *green, blue* and *red* boxes. (B) Plot of spine density versus age for dendritic segments visualized in live slices. (C) Cumulative histogram showing distribution of spine lengths for P6 – P11 (*green*) and P14 – P25 (*purple*) mice. (D) Distribution of spines (indicated by *vertical lines*) along continuous stretches of dendrite from 20 different cells. (E) Plot of spine density versus distance from the 20 dendrites in previous panel (*gray dashed lines*) and averages with s.e.m. (*white boxes*).

## Results

### Density and morphology of dendritic spines in live and fixed SNc dopaminergic neurons

We analyzed the density and morphology of dendritic spines on dopaminergic neurons in live brain slices obtained from juvenile (P6–25) mice. *Figure 1* shows a typical example of an Alexa-594-filled dopamine neuron. Dendritic spines were clearly visible on all dendrites that were imaged, including on proximal and distal dendrites (*Figure 1A*). However, the density of spines varied widely across the population of dendritic segments from 0.5 spines/10 μm up to 4.4 spines/10 μm, with the average density on segments at 2.08 ± 0.10 spines/10 μm (n = 76; *Figure 1B*). Plotting the spine density versus age (*Figure 1B*), we observed a weak but statistically significant correlation indicating a reduction in density with age (Pearson's R = -0.23569, p=0.040, n = 76 dendritic segments).

Analysis of spine morphology showed many of the typical spine shapes (e.g. stubby and mushroom spines) similar to those reported on more traditional spiny neurons like pyramidal cells. In addition, we observed many spines that exhibited strikingly long necks, characteristic of dendritic filopodia (*Figure 1C*). In young mice (P6–P11), the average spine length was 1.97 ± 0.06 μm (range, 0.18–4.85 μm) with 142 of 355 spines (40%) measuring >2 μm in length (*Figure 1C*). In older juveniles (P14–P25), spines were somewhat shorter at an average length of 1.74 ± 0.05 μm (range, 0.06–4.95 μm) with 166 of 518 spines (32%) measuring >2 μm. Therefore, we observe a significant shift in the density of spines during development away from long spines toward shorter spines, consistent with the notion that longer, filopodial-like spines may be immature structures.

In separate experiments, we analyzed the density of spines across the entire visible dendritic tree in individual cells (age P14–P18). In the 'whole cell' images, the average spine density was 1.86 ± 0.02 spines/10 μm (n = 20; *Figure 1D,E*). *Figure 1D* shows the positions of individual spines within a continuous stretch of dendrite for 20 neurons. In some cells, spines were evenly distributed throughout the dendrites while in others, we observed stretches of dendrites that were only sparsely populated with spines. This fits with the large variability in spine density that we observe between individual dendritic segments (*Figure 1B*), as well as observations in primates that spines can occur in patches (*Schwyn and Fox, 1974*). On average, the density of spines decreased modestly with distance from 2.33 ± 0.30 spines/10 μm (n = 20) in the proximal dendrite (0–100 μm) to 1.52 ± 0.22 spines/10 μm (n = 12) in the most distal dendrites (300–545 μm) (p=0.037, student's unpaired). This is also consistent with results in guinea pig (*Yung et al., 1991*) and humans (*Cruz-Sanchez et al., 1995*) that describe a higher density of spines in proximal versus distal dendrites.

To test the possibility that the dendritic spines identified in live SNc dopamine neurons may be an artifact of the slicing procedure (*Kirov et al., 1999*), we imaged dopamine neurons in both Golgi-stained tissue and fast-perfusion fixed tissue. In Golgi-stained tissue (*Figure 2A*), SNc dopamine neurons exhibited dendritic spines consistent with past experiments analyzing Golgi-stained SNc cells (*Juraska et al., 1977*; *Phelps and Adinolfi, 1982*; *Rinvik and Grofova, 1970*; *Schwyn and Fox, 1974*). To visualize dopamine neurons in tissue from fast transcardially-perfused mice, we used transgenic mice in which GFP is driven by the tyrosine hydroxylase promoter and visualized their morphology using juxtacellular labeling with Alexa-594. Spines in fixed tissue were observed along the dendrites of SNc dopamine neurons, though at somewhat lower densities than in live tissue (*Figure 2B*). This raises the possibility that some spine growth in dopamine neurons may result from acute slicing. Alternatively, it has been suggested that tissue fixation may limit diffusion of fluorescent dye and visualization of spines (*Kim et al., 2007*). Comparing cells from young (P15–P25) and mature (P75–P119) mice (*Figure 2C*), we found no significant difference in the density of spines (young, 0.90 ± 0.15 spines/10 μm, n = 16; mature, 0.75 ± 0.13 spines/10 μm, n = 13; p=0.47).

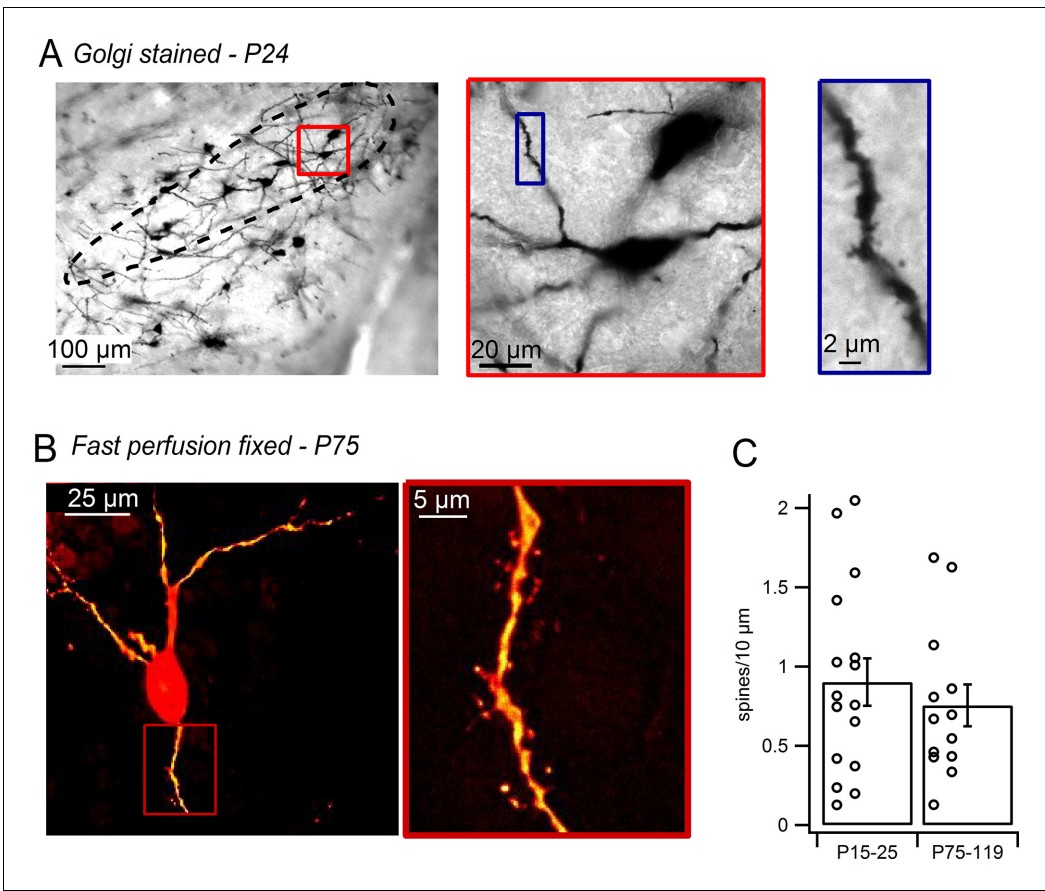

**Figure 2.** Dendritic spines on SNc dopamine neurons visualized in perfusion-fixed slices. (**A**) Golgi stained sagittal slice from P24 mouse. SNc indicated by *black outline*. *Red box* shows a putative dopamine neuron at higher magnification. *Blue box* shows selected dendritic segment with dendritic spines. (**B**) SNc dopamine neuron from brain of transcardially-perfused, P75 mouse visualized by juxtacellular labeling with Alexa-594. *Red box* shows spiny dendritic segment at higher magnification. (**C**) Bar plot of average SNc dopamine neuron spine densities measured in perfusion-fixed brain slices.

Therefore, we demonstrate that dendritic spines are present in live and fixed tissue preparations from both juvenile and adult mice, suggesting that dendritic spines are a common feature of mammalian SNc dopamine neurons.

## Ca²⁺ imaging reveals active sites of synaptic release at spines on SNc neurons

We next tested whether dendritic spines on SNc dopamine neurons are sites of glutamatergic synaptic input. Using locally-positioned theta-glass electrodes (diameter, 5–10 µm), we tested whether electrical stimulation would generate localized $Ca^{2+}$ influx into the spine head indicating the presence of active presynaptic inputs (*Figure 3A*) (*Chalifoux and Carter, 2010*; *Oertner et al., 2002*; *Sabatini et al., 2002*). Dopamine neurons were filled via patch pipette with Alexa-594 and Fluo5F to visualize cell morphology and intracellular $Ca^{2+}$. To eliminate spontaneous firing and $Ca^{2+}$-dependent oscillations (*Nedergaard et al., 1993*; *Puopolo et al., 2007*; *Wilson and Callaway, 2000*), we added QX-314 (1 mM) to the pipette solution and nifedipine (10 µM) to bath solutions to block voltage-gated sodium and calcium channels.

*Figure 3B* provides an example of $Ca^{2+}$ signals recorded in two adjacent spines and the neighboring dendrite in response to electrical stimulation. Following stimulation, we observed a large, rapid increase in the $Ca^{2+}$ signal in spine 1 (*green trace*), but not in spine 2 (*black trace*) or the nearby dendrite (*blue trace*). Comparing the amplitude and rise times of $Ca^{2+}$ signals in spines

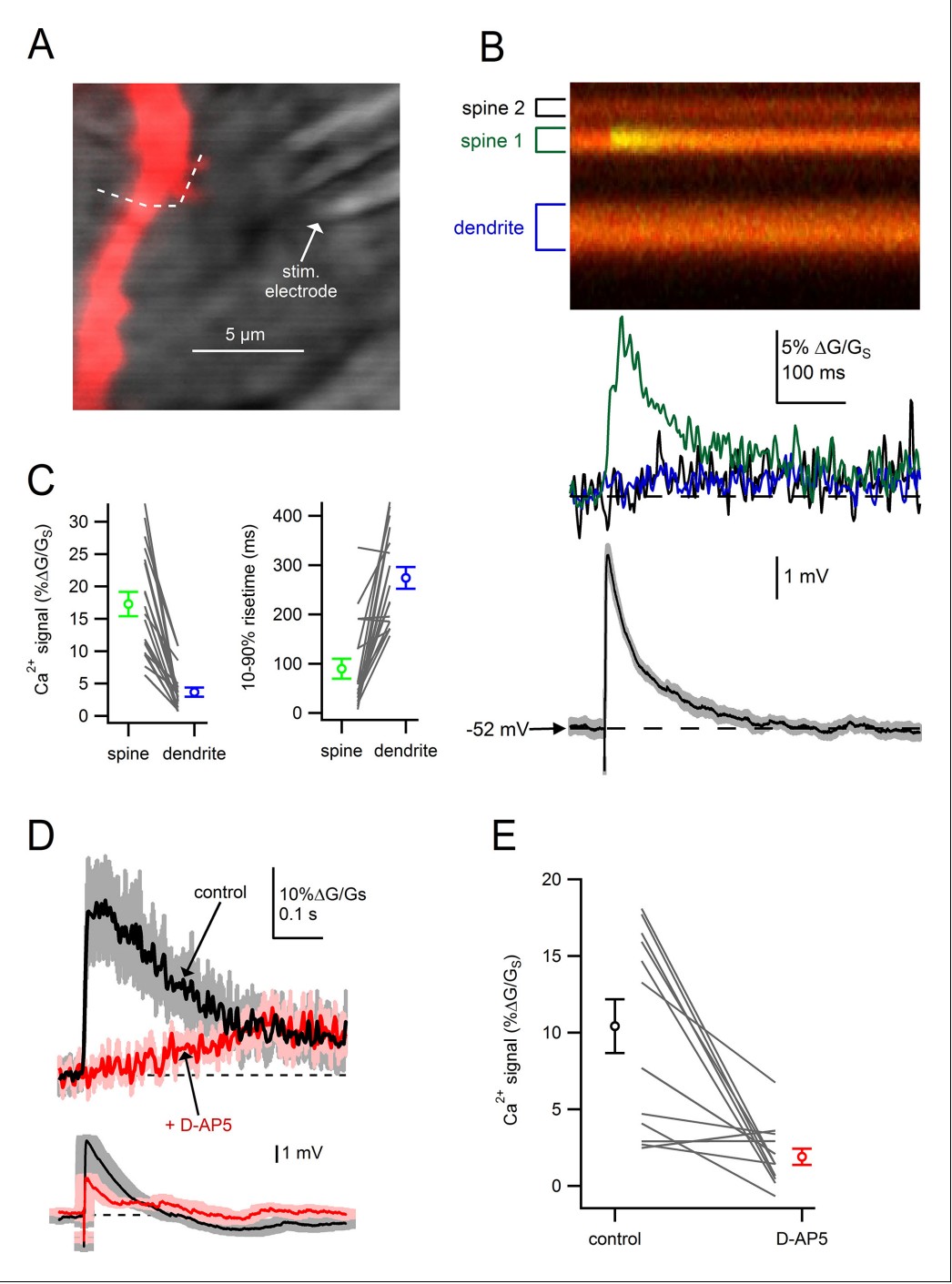

**Figure 3.** Dendritic spines on SNc dopamine neurons are sites of glutamatergic synaptic inputs. (**A**) Dendritic segment visualized with Alexa-594 and Dodt contrast image to visualize the stimulation electrode. *White dashed line* indicates path of linescan. (**B**) Linescan images of Alexa-594 (*red*) and Fluo5F (*green*) and quantified Ca$^{2+}$ signals in spine 1 (*green*), spine 2 (*black*) and the dendrite (*blue*). Simultaneously recorded somatic EPSP shown below. (**C**) Peak amplitudes and rise times of synaptically-evoked Ca$^{2+}$ signals into a spine and parent dendrite for all spines tested (*black lines*). Average values and s.e.m. for spine (*green*) and dendrite (*blue*) shown in outer circles. (**D**) *top:* Ca$^{2+}$ influx into the spine head in response to synaptic stimulation in control conditions (*black*) and after wash on of 50 µM D-AP5 (*red*). *bottom*: Corresponding somatically recorded EPSPs. (**E**) Peak amplitude of spine Ca$^{2+}$ signal before and after application of D-AP5.

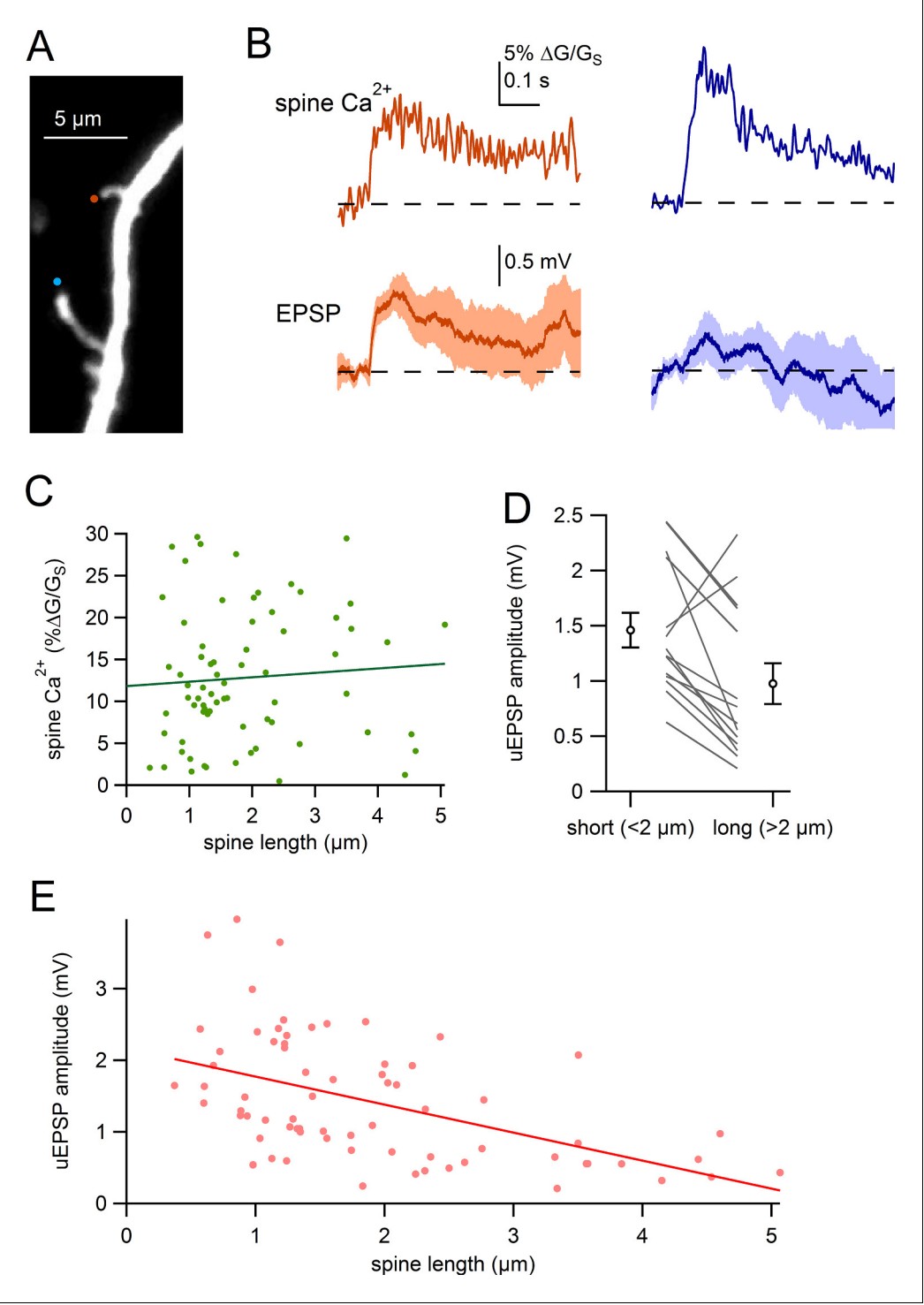

**Figure 4.** Comparison of uEPSPs for short and long spines. (**A**) Dendritic segment with neighboring spines of different lengths visualized by Alexa-594 (bottom spine = 4.2 μm, top spine = 1.6 μm). Sites of glutamate uncaging are indicated by *blue and orange circles*. (**B**) Glutamate uncaging-evoked responses of spines in A. *Blue traces* correspond to the long spine, *orange traces* correspond to the short spine. *Top*: glutamate-evoked spine $Ca^{2+}$ signals. *Bottom*: uEPSPs. (**C**) Amplitude of spine $Ca^{2+}$ signal plotted against spine length for all spines tested (*dots*) and linear regression. (**D**) uEPSP amplitudes of neighboring short (<2 μm) and long (>2 μm) spines. Circles represent mean and s.e.m. (**E**) Amplitudes of uncaging-evoked EPSPs plotted against spine length for all spines tested (*dots*) and linear regression.

*Figure 4 continued on next page*

*Figure 4 continued*

The following figure supplement is available for figure 4:

**Figure supplement 1.** Glutamate uncaging-evoked responses of dendritic spines.

versus dendrites, $Ca^{2+}$ signals in individual spine heads were larger (p=3.4e-7) and faster (p=6.3e-7, student's paired t-test; n = 19) than in the nearby dendrite (*Figure 3C*). Finally, we tested the effect of NMDA receptor blockade on synaptically-evoked $Ca^{2+}$ signals. Across all spines tested, application D(-)-2-amino-5-phosphonopentanoic acid (D-AP5; 50 μM) produced a dramatic reduction in $Ca^{2+}$ influx into the spine (*Figure 3D,E*; control: 10.1 ± 1.9% $\Delta G/G_S$; D-AP5: 2.0 ± 0.6% $\Delta G/G_S$; p=0.9e-4, students paired t-test, n = 12), suggesting that NMDA receptors are the main source of $Ca^{2+}$ entering spines. This observation is consistent with the spine as a site of glutamatergic synaptic input innervated by active presynaptic terminals.

## Electrical and chemical compartmentalization in dendritic spines of SNc dopamine neurons

The presence of glutamatergic synapses on both dendritic shafts and spines in dopamine neurons raises the possibility that these two structural classes of synapses could differ in chemical and electrical signaling. Chemical compartmentalization by the spine, due to slower diffusion through the narrow spine neck, could result in greater spatial-specificity of cell signaling pathways in the spine head. Similarly, spine and shaft synapses could differ in synaptic strength due to attenuation of the EPSP by the resistance of the spine neck. Finally, the spine neck could produce local boosting of the EPSP within the spine head leading to activation of voltage-dependent ion channels that would not take place at shaft synapses.

Differences in the geometry between individual spines are thought to shape chemical and electrical signaling (*Araya et al., 2006*; *Bloodgood and Sabatini, 2005*; *Noguchi et al., 2005*; *Takasaki and Sabatini, 2014*). For example, work in cortical pyramidal neurons has shown a clear inverse correlation between synaptic potential and neck length (*Araya et al., 2014*) while similar work in hippocampal pyramidal neurons show only a weak relationship (*Takasaki and Sabatini, 2014*). Therefore, we tested electrical signaling in single spines of different lengths using glutamate uncaging. Much like the experiments examining synaptically-evoked responses, glutamate uncaging resulted in uncaging-evoked EPSPs (uEPSPs), clear $Ca^{2+}$ influx into the spine head, and a smaller, slowly rising $Ca^{2+}$ signal in the parent dendrite (*Figure 4— figure supplement 1*).

*Figure 4A* shows an example of two neighboring spines of different lengths and their uncaging-evoked responses. Using the same uncaging power to activate each spine, we found that the two spines display comparable uncaging-evoked $Ca^{2+}$ signals (*Figure 4B*, *top*). However, we found clear differences in the amplitudes of uEPSPs. Glutamate uncaging generated a uEPSP of 0.91 mV in the shorter spine (1.55 μm) versus a uEPSP of 0.32 mV in the longer spine (4.15 μm). In collected experiments, we observed no correlation between the amplitude of the spine $Ca^{2+}$ signal and spine length (*Figure 4C*; Pearson's R = 0.074; p=0.54, n = 71). However, comparing uEPSP amplitudes for neighboring spines, uEPSPs from short spines (<2 μm) were significantly larger than uEPSPs generated in nearby longer spines (>2 μm) (short: 1.46 ± 0.16 mV versus long: 0.98 ± 0.19 mV; p=9.8e-3, paired, n = 14) (*Figure 4D*). Furthermore, plotting the amplitude of uEPSPs against spine length for all spines assayed (*Figure 4E*), we observed a strong correlation between the size of the somatically-recorded uEPSP and spine length (slope = -0.39 mV/μm; Pearson's R = -0.50; p=1.5e-5, n = 71). In agreement with *Araya et al. (2006)*, our data demonstrate that synapses formed onto long spines result in smaller amplitude uEPSPs, and likely exert a weaker influence on the membrane potential in SNc dopamine neurons.

Because spines can compartmentalize chemical signals due to their thin spine neck which acts as diffusion barrier, we next analyzed the influence of spine length on chemical compartmentalization. We measured the rate of fluorescence recovery after photobleaching (FRAP) in spines of varying lengths (0.88–4.54 μm). As seen in *Figure 5A and B*, fluorescence intensity of Alexa-594 in the spine head was measured by linescans across the spine, while a second two-photon laser was used to partially bleach the dye within the spine (0.5 ms pulse, 725 nm). Traces were corrected for a small

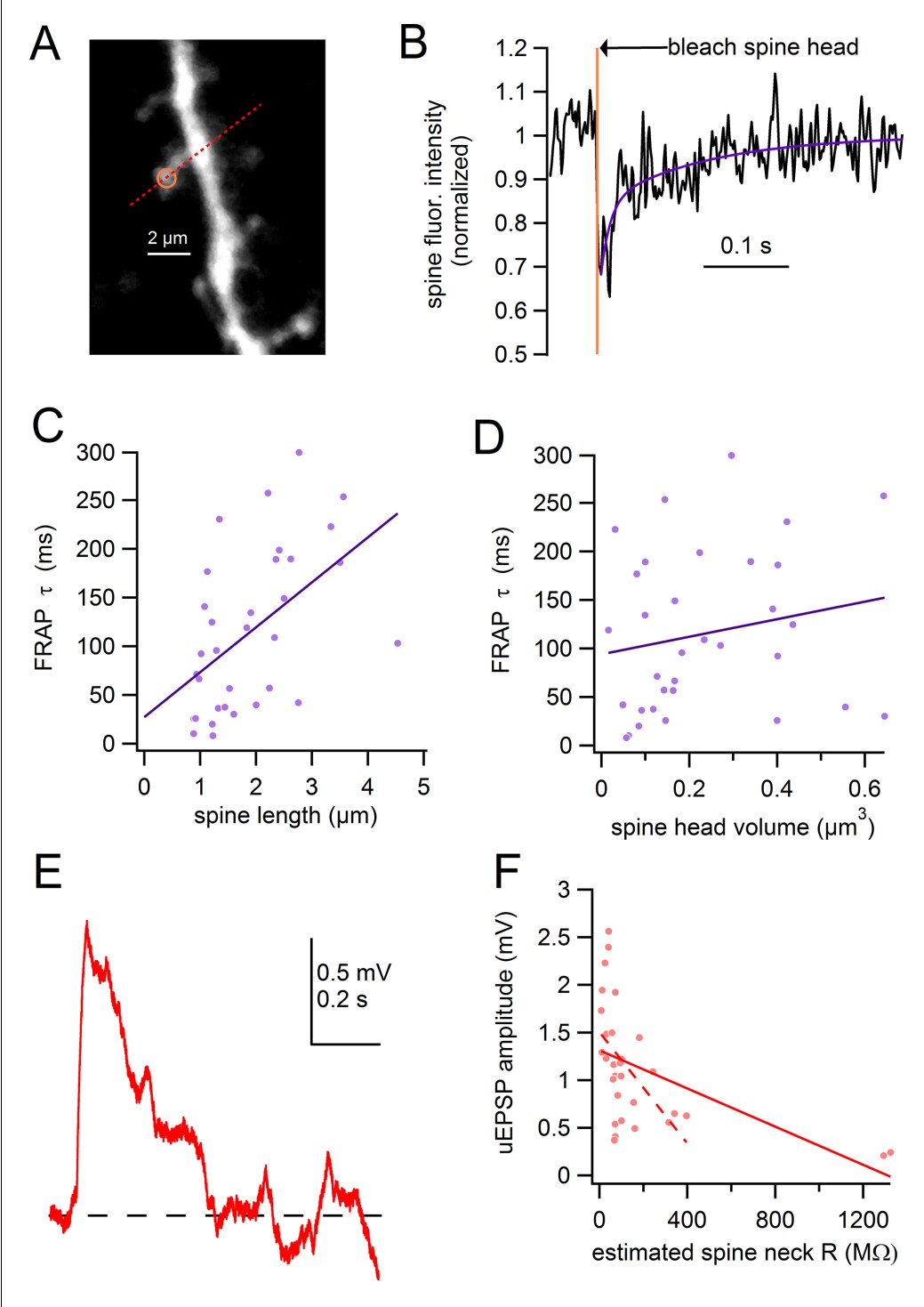

**Figure 5.** Spine length correlates with chemical compartmentalization. (A) Frame scan of assayed spine, path of linescan (*red line*), and site of photobleaching (*yellow*). (B) Spine fluorescence intensity following selective photobleaching of spine head. Timing of photobleaching pulse indicated by *vertical yellow line*. Subsequent recovery of fluorescence was fit with a double-exponential function (*purple*). (C) Time constant of FRAP plotted against spine length with linear regression (*purple line*). (D) Time constant of FRAP plotted against spine head volume with linear regression (*purple line*). (E) uEPSP evoked from spine shown in A. (F) uEPSP amplitude plotted against estimated spine neck resistance with linear regression for all data (*solid red line*). Linear regression for all data except the 2 spines with the highest estimated neck resistance indicated by *dashed line*.

amount of photobleaching (~3.5%) which occurred as a result of scanning by the imaging laser alone (see *Materials and methods*). We then fit the recovery of spine fluorescence intensity with a double-exponential function (*Figure 5B*, *purple line,* weighted tau = 66.5 ms) and plotted the weighted time constant of FRAP against spine length for 33 spines (*Figure 5C*). We observed a significant correlation in which the time course of FRAP slowed with increased spine lengths (slope = 46.1 ms/μm; Pearson's R = 0.52; p=0.0020). The range of FRAP time courses observed (8.1–299 ms, mean = 115 ms, median = 103 ms) were similar to analogous experiments conducted in hippocampal pyramidal neurons (*Grunditz et al., 2008*; *Takasaki and Sabatini, 2014*; *Tonnesen et al., 2014*). These FRAP experiments demonstrate that spine length significantly influences the compartmentalization of chemical signals in SNc dopamine neurons.

Previous work has shown that the dimensions of the spine head can influence the time course of the FRAP measurements (*Svoboda et al., 1996*; *Tonnesen et al., 2014*). Therefore, we estimated the volume of the spine head based on the full width at half maximum (FWHM) of Alexa-594 signal intensity across the spine head. Assuming that the spine head is a sphere with the FWHM as the diameter, we estimated that spine head volumes cover a range from 0.016 to 0.64 μm$^3$. We found no correlation between the FRAP time constants and spine head volume (Pearson's R = 0.193, p=0.28). We next used estimates of the spine volume and FRAP measurements to calculate the resistance of the spine neck with Fick's law: $R_{neck}=tau_{FRAP}D_{Alexa}R_{axial}/V_{head}$. In this equation, $R_{neck}$ is the neck resistance, $tau_{FRAP}$ is the time constant from FRAP measurements, $D_{Alexa}$ is the free diffusion time constant for Alexa-594 (120 μm$^2$/s), $R_{axial}$ is the axial resistance (150 Ω*cm), and $V_{head}$ is the volume of the spine head. Using these values, the estimated spine neck resistance ranged from 8.4 MΩ–1.3 GΩ, with a median of 72 MΩ. These estimates assume a spherical spine head as well as an intermediate value of cytoplasmic $R_{axial}$ of 150 Ω*cm, while values can range from 50–250 Ω*cm. Despite the assumptions, however, our neck resistance values fall roughly in line with estimates from studies in other cell types (*Tonnesen et al., 2014*).

Given the estimates of $R_{neck}$ that we obtained using the spine morphology and FRAP measurements, we compared these measurements with the size of evoked synaptic potentials. Therefore, we uncaged glutamate onto the same spines from which FRAP analysis was performed and measured the uEPSPs (*Figure 5E*). Plotting the uEPSP amplitude against the $R_{neck}$, we found a moderate inverse correlation between uEPSP amplitude and $R_{neck}$ (Pearson's R = -0.503, p=0.0046) (*Figure 5F*). Intriguingly, the 2 spines which produced the smallest amplitude uEPSPs also had estimated neck resistances that were dramatically greater than the other spines assayed. A significant correlation is still observed if these data points are excluded from analysis (*Figure 5F dashed line*; R = -0.488, p=0.0085). In summary, these findings are consistent with the idea that the higher resistance of longer spines may result in stronger attenuation of the EPSP across the spine neck (*Araya et al., 2006*; *2014*).

## NMDA and AMPA-receptor mediated conductances in dopamine neuron spines

In addition to spine geometry, synaptic receptor composition such as the density of spine AMPA receptors can be an effective determinant of synaptic strength (*Matsuzaki et al., 2001*). Comparisons of uEPSPs kinetics in long and short spines indirectly support this idea. We found significantly faster rise times in short spines than in longer spines (10–90% uEPSP rise time; short <2 μm spines: 42.8 ± 6.5 ms, long ≥ 2 μm spines: 71.9 ± 10.4 ms; n = 45 and 26; p=0.023 student's unpaired t-test). Therefore, we hypothesized that the variability in the uEPSP amplitudes may result from distinct compositions of synaptic glutamate receptors in the short and long spines. To test this, we recorded glutamate uncaging-evoked AMPA or NMDA receptor-mediated currents from short and long-necked spines by voltage clamping neurons to either −70 mV or +40 mV (*Figure 6A,B*). To maximize space clamp, we analyzed spines located within 50 μm of the soma and recorded using Cs$^+$-based internal solutions and bath applied TTX (500 nM). NMDAR-mediated currents were quantified 100 ms following onset of the uncaging pulse when AMPA receptors were likely desensitized.

The uncaging-evoked AMPAR-mediated current measured at −70 mV was dramatically larger in short spines versus longer spines (slope = 4.24 pA/μm; Pearson's R = -0.60, p=2.9e-6, n = 52 spines) (*Figure 6C*). Holding at +40 mV to measure NMDAR-mediated currents, again we found a significant, yet somewhat weaker negative relationship between the uncaging-evoked NMDAR current and spine length (slope = -2.3 pA/μm; Pearson's R = -0.49; p=2.3e-4, n = 52 spines) (*Figure 6C*).

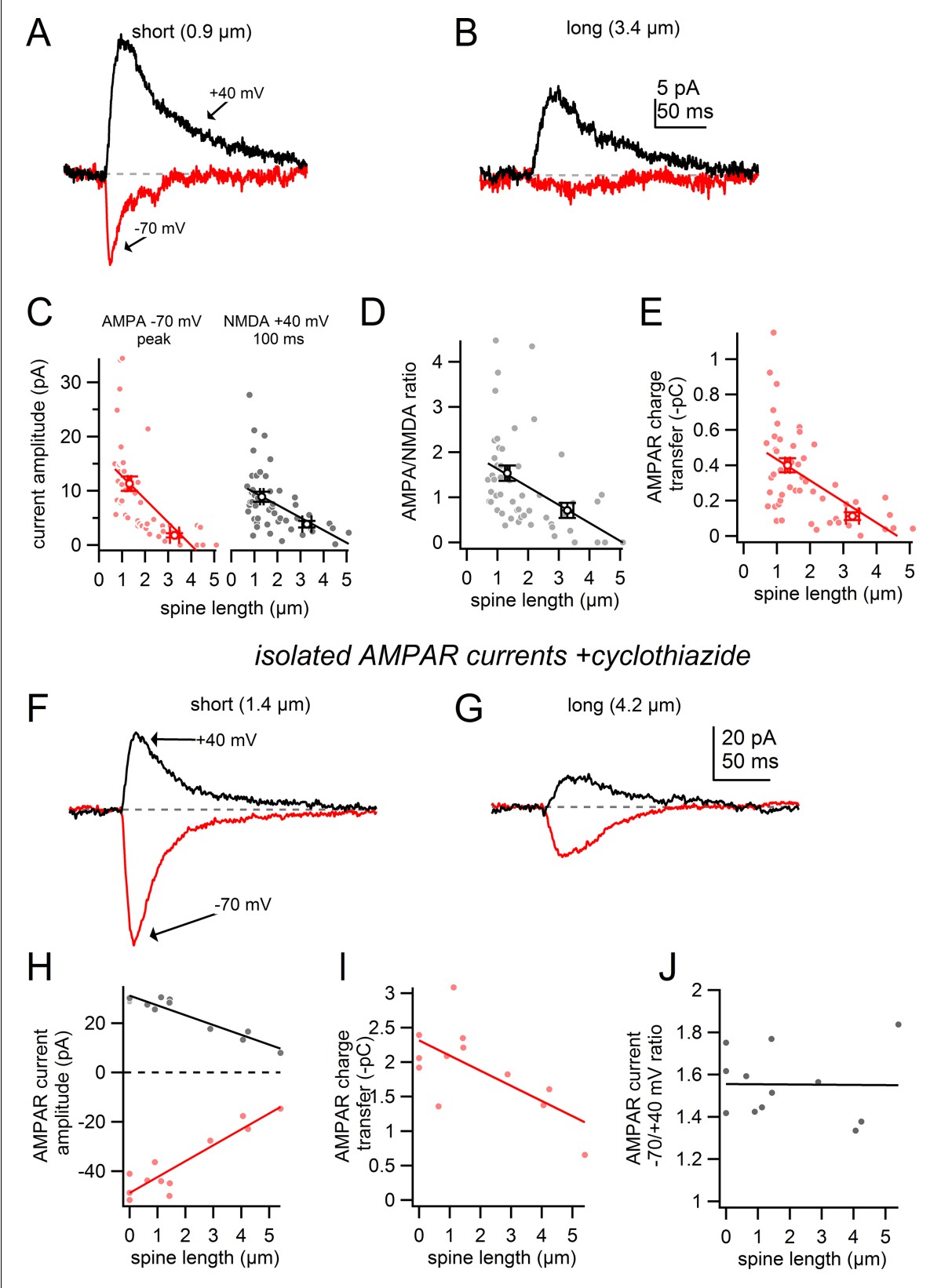

**Figure 6.** Comparison of AMPA/NMDA ratio for short and long spines. (**A**) Uncaging-evoked currents recorded at -70 mV (*red*) and +40 mV (*black*) from targeting a short spine (0.9 μm). (**B**) Uncaging-evoked currents as in A from targeting a nearby long spine (3.4 μm). (**C**) Plot of AMPAR (*red*) and NMDAR

*Figure 6 continued on next page*

Figure 6 continued

(*black*) – mediated current amplitudes versus spine length for all spines tested and corresponding linear regressions. (**D**) Plot of AMPA/NMDA ratio versus spine length for all spines tested and linear regression. *Circles* indicate mean and s.e.m. for short and long spines. (**E**) Plot of AMPAR charge transfer versus spine length for all spines test and linear regression. (**F**) Uncaging-evoked currents as in **A** in the presence of CTZ and D-AP5 while targeting a short spine (1.4 μm). (**G**) Uncaging-evoked currents as in **F** from targeting a nearby long spine (4.2 μm). (**H**) Plot of AMPAR– mediated current amplitudes measured at −70 mV and +40 mV in the presence of CTZ and corresponding linear regressions. (**I**) Plot of AMPAR charge transfer versus spine length and linear regression. (**J**) Plot of the ratio of AMPAR current amplitudes measured at -70 and +40 mV versus spine length and linear regression.

Lastly, we found that the AMPA to NMDA ratio was significantly lower in long spines (>2 μm) as compared to short spines (spine length <2 μm, AMPA/NMDA ratio = 1.54 ± 0.17; spine length >2 μm, AMPA/NMDA ratio = 0.71 ± 0.17; p=0.0017, n = 52 spines) (*Figure 6D*). In addition to spine neck resistance, these findings suggest that the negative correlation between uEPSP amplitude and spine length (*Figure 4E*) also involves a differential glutamate receptor expression and composition, with a weaker AMPA-receptor component present in long-necked spines.

The spine neck imparts an electrical resistance, and therefore may attenuate synaptic current entering the dendrite. The extent of this attenuation will depend on the relative values of the synaptic conductance and spine neck resistance. Faster currents (such as those mediated by AMPARs) will likely display greater attenuation than slower NMDAR-mediated currents (*Johnston and Wu, 1995*). Therefore, it is possible that the observed difference in AMPA to NMDA ratio is due to greater filtering of AMPAR currents by the spine neck resistance of long spines. Measurements of synaptic charge transfer are less distorted by the cable properties of the neuron and spine neck than are measurements of current amplitude (*Johnston and Wu, 1995*).Therefore, we examined the relationship between the AMPAR-mediated charge transfer (measured as the integral of the uncaging-evoked current at −70 mV) and the length of the spines (*Figure 6E*). We observed a moderate and highly significant correlation between these values (R = −0.56, p=1.8e-5, n = 52 spines), further supporting the hypothesis that long spines are of weaker strength due in part to a smaller contribution of AMPARs.

To further disambiguate whether the relationship between AMPAR current amplitude and spine length was due to differences in receptor expression or differences in spine neck resistance, we examined the relationship between isolated AMPAR-currents (NMDARs were blocked with 50 μM D-AP5) and spine length in the presence of cyclothiazide (CTZ, 100 μM) to inhibit AMPAR desensitization. CTZ increased the maximum synaptic conductance and slowed the time course of the AMPAR-mediated conductance changes. Both effects will minimize the impact of the spine neck resistance on measurement of the synaptic current. Examples of uncaging-evoked AMPAR currents at positive and negative potentials in the presence of CTZ are shown in *Figure 6F,G*. We observed a significant correlation between AMPAR current amplitude and spine length in the presence of CTZ at both positive and negative voltages (at -70 mV R = 0.92, p=2.5e-5; at +40 mV R = 0.95, p<1e-5; n = 6 spines, 3 shafts) (*Figure 6H*). We also observed a significant correlation between AMPAR-mediated charge transfer and spine length in the presence of CTZ (R = 0.66, p=0.02, n = 6 spines, 3 shafts) (*Figure 6I*) further supporting the hypothesis that long spines display lower AMPAR density.

Finally, we confirmed that the spine neck acts as an ohmic resistor. If the spine neck was non-ohmic (i.e. voltage-dependent) it could differentially affect measurement of the AMPA/NMDA ratio in spines with different neck resistances. Measuring isolated AMPAR currents at different voltages allowed us to test the ohmic nature of the spine neck resistance because changes in synaptic conductance will be due the presence of the ligand alone. To measure the synaptic conductance in the absence of any spine neck resistance, we uncaged glutamate directly near the dendritic shaft. When holding the cell at −70 mV and targeting the dendritic shaft, uncaging-evoked synaptic currents were 1.60 ± 0.05 times larger than the current measured at +40 mV (*Figure 6J*). If the spine neck resistance changed with membrane potential, one would predict that the ratio between AMPAR currents measured at −70 mV and +40 mV would differ according to spine neck resistance. However, when uncaging onto spines the ratio of AMPAR current amplitudes measured at −70 and +40 mV was 1.56 ± 0.05 (*Figure 6J*) - similar to trials targeting dendritic shafts. Furthermore, we observe no correlation between spine length and the ratio of AMPAR currents measured at −70 and +40 mV

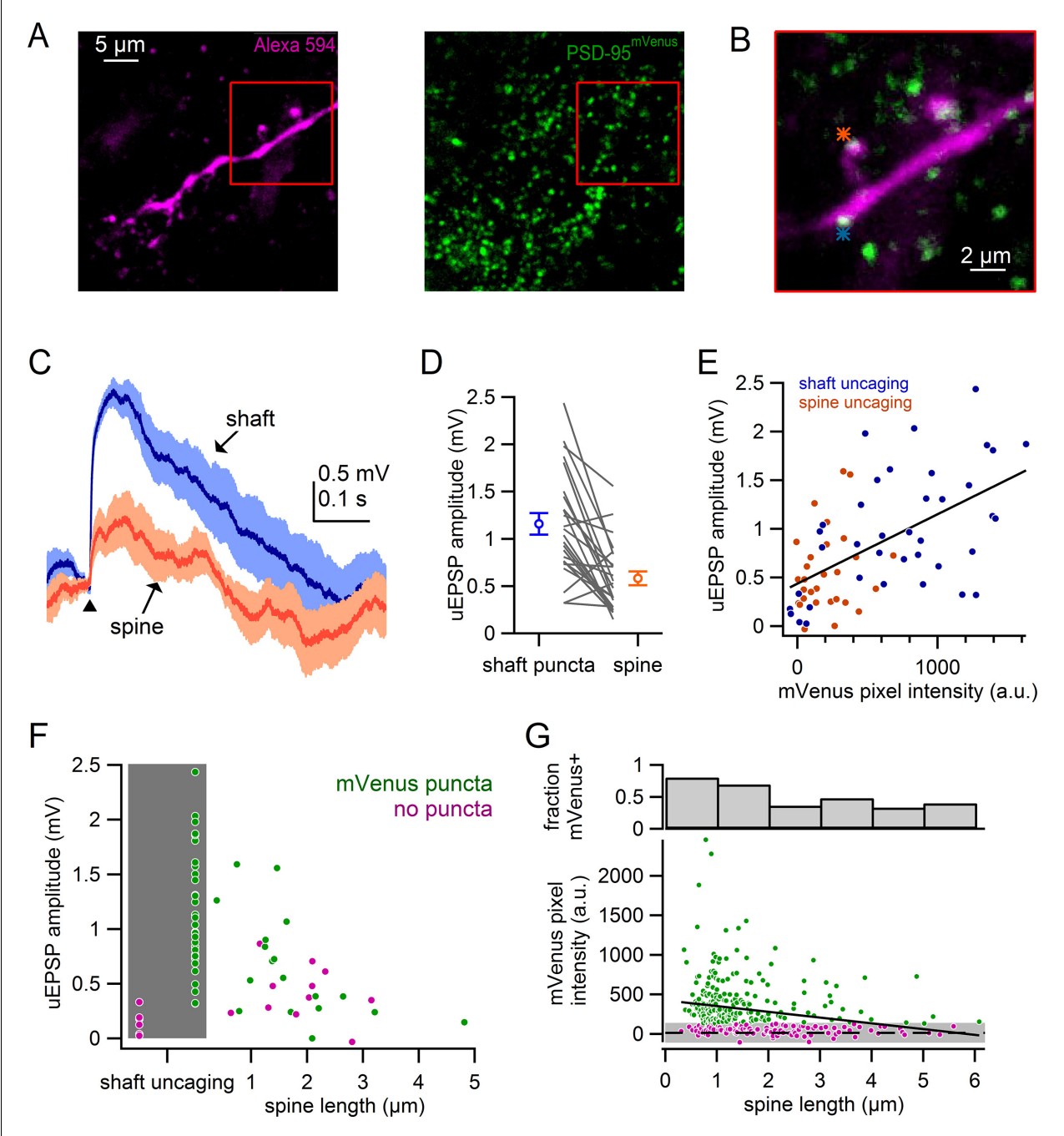

**Figure 7.** Comparison of spine and shaft synapses. (A) Maximum intensity projections of two photon images from a recorded neuron in DAT-Cre x PSD-95-ENABLED mouse.*Left*: Alexa-594-filled dendrite. *Right*: PSD-95^mVenus puncta. *Red boxes* indicate region shown in B. (B) Overlay of a single slice of the z-stack from images in A, *asterisks* indicate sites of glutamate uncaging targeting a spine (*orange*) and a shaft synapse (*blue*). (C) uEPSPs evoked in response to targeting of a shaft puncta (*blue*) and spine (*orange*) located on the same dendrite. (D) Comparison of uEPSP amplitudes when targeting shaft synapses or nearby spines. Circles represent averages and s.e.m. (E) Plot of uEPSP amplitude versus mVenus pixel intensity for shafts (*blue*) and spines (*orange*). (F) Plot of uEPSP amplitude versus spine length with shaft uncaging sites indicated by *grey box*. *Green circles* indicate that mVenus pixel intensity was >2 standard deviations above background green pixel intensity. *Magenta circles* indicate that the spine or shaft did not display significant mVenus signal. (G) Plot of mVenus pixel intensity and the fraction of mVenus-positive spines against spine length. *Grey box* indicates average background mVenus signal ± 2 standard deviations. *Green* and *magenta* circles indicate spines with or without significant mVenus signal.

The following figure supplement is available for figure 7:

*Figure 7 continued on next page*

*Figure 7 continued*

**Figure supplement 1.** Plot of the measured spine density versus the measured shaft synapse density for individual dendritic segments (filled circles) and average values (empty circle).

(R = -0.010, p=0.97,n = 3 shafts, 6 spines) (*Figure 6J*). Therefore, it appears that the resistance of the spine neck does not change with membrane potential under our experimental conditions.

## Comparison of uncaging responses in spines and PSD95-positive shaft synapses

Glutamatergic excitation of SNc dopamine neurons has long been recognized to occur via input to synapses located on the dendritic shaft (*Chatha et al., 2000*; *Henny et al., 2012*; *Paquet et al., 1997*; *Rinvik and Grofova, 1970*). However, it remains unclear how spine and shaft synapses compare in both density and function. The difficulty in identifying shaft synapses in live slices has limited functional investigation of shaft synapses. To address this limitation, we utilized a recently developed transgenic mouse in which postsynaptic protein PSD-95 is tagged with mVenus (ENABLED) to allow unambiguous identification of shaft synapses (*Fortin et al., 2014*). Crossing the ENABLED mouse with a dopamine transporter driven Cre mouse line (DAT-Cre), resulted in dopamine neuron-specific expression of the postsynaptic protein PSD-95 tagged with mVenus.

*Figure 7A* shows an example dendritic segment filled with Alexa-594, in which postsynaptic densities are identified as mVenus-positive puncta. We compared activation of spine synapses with puncta-labeled synapses on the neighboring dendritic shaft (*Figure 7B*). Uncaging onto shaft synapses resulted in uEPSPs that were 100% larger than uEPSPs from nearby spine synapses (shaft: 1.16 ± 0.11 mV; spine: 0.58 ± 0.07 mV; n = 24, p=2.7e-5, student's paired t-test) (*Figure 7C,D*). By contrast, uncaging onto mVenus-negative dendritic regions resulted in uEPSPs that were minimal in amplitude (puncta: 1.40 ± 0.15 mV, no puncta: 0.14 ± 0.05 mV; n = 6, p=4.9e-4, student's paired t-test). These results confirm that mVenus puncta clearly identify shaft synapses.

The presence of PSD-95 has been shown to stabilize AMPARs within the synapse (*Béïque et al., 2006*; *El-Husseinl et al., 2000*; *Taft and Turrigiano, 2014*), which is expected to correlate with stronger postsynaptic potentials. We next compared the pixel intensity of mVenus at the position of uncaging with uEPSP amplitudes (*Figure 7E*). We observed a significant correlation between uEPSP amplitude and mVenus pixel intensity (Pearson's R = 0.57, n = 67, p=4.8e-7). Regions of the dendrites, including the dendritic shaft and spines, were considered mVenus-positive when the green pixel intensity exceeded 2 standard deviations of the average green background intensity measured in the dendrite shaft. It is possible that small mVenus puncta could escape our detection, biasing our results to synapses with large postsynaptic densities. Using this criteria, we compared spines of similar lengths (<2 μm) and found that uEPSPs were substantially larger for PSD-95 positive spines than for PSD-95 negative spines (*Figure 7F*) (PSD-95 positive uEPSP = 0.85 ± 0.13 mV, n = 12; PSD-95 negative uEPSP=0.42 ± 0.12 mV, n = 5; p=0.030). These results show that in both shaft and spine synapses, synaptic strength correlates with PSD-95 expression.

Given our observations that synaptic strength is correlated with both spine length and PSD-95 expression, we next asked whether PSD-95 expression differs in spines of various lengths. We measured mVenus pixel intensity in 327 spines from 11 dopamine neurons and plotted mVenus pixel intensity against spine length (*Figure 7G*). We observed a weak, but significant correlation between mVenus expression and spine length (R = −0.23, p=2.5e-5). We again categorized spines as mVenus positive if pixel intensity was more than 2 standard deviations greater than the background signal. 75% of spines with lengths <2 μm (179 of 240) were scored mVenus positive whereas only 40% of spines longer than 2 μm (33 of 83) displayed detectable mVenus expression. This observation is consistent with the conclusion from our voltage-clamp analyses, that longer spines display smaller AMPAR currents (*Figure 6*), likely contributing to the smaller uEPSPs observed in current clamp (*Figure 4*). In total, our data support the hypothesis that the long spines are immature synaptic structures that have few AMPARs.

Lastly, we compared the relative densities of spines and shaft synapses on individual dendritic segments (*Figure 7—figure supplement 1*). In most cases, mVenus-positive shaft synapses outnumbered spines (average density shaft synapses: 2.53 ± 0.14 puncta/10 μm; average density spine

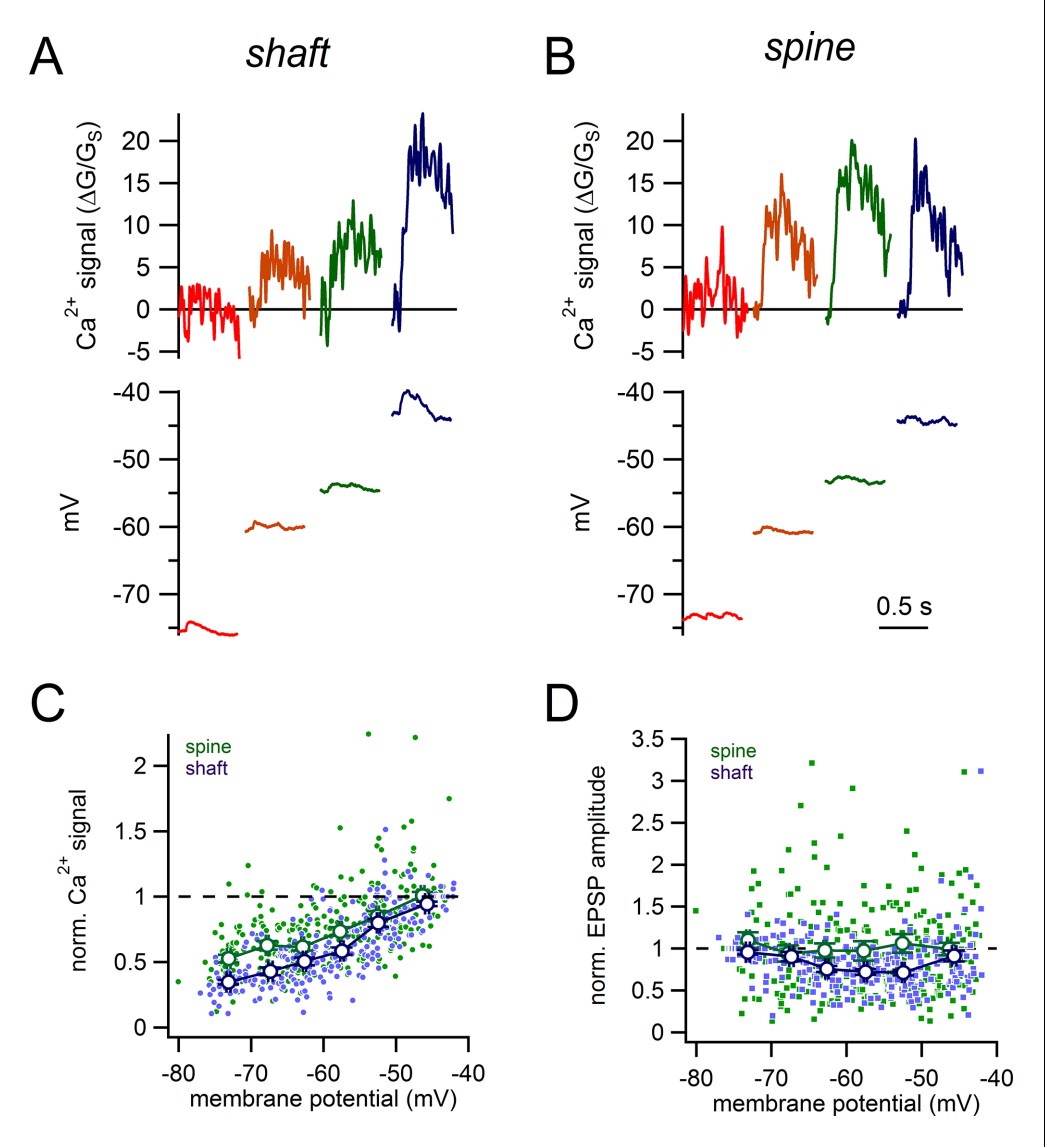

**Figure 8.** Voltage-dependence of shaft and spine synaptic responses. (A) Examples of glutamate-evoked $Ca^{2+}$ signals and uEPSPs evoked from 4 different stable membrane potentials for a single shaft synapse. (B) Examples of glutamate-evoked $Ca^{2+}$ signals and uEPSPs evoked from 4 different stable membrane potentials for a single spine. (C) Plot of normalized $Ca^{2+}$ signal amplitudes against starting membrane potential for spines (*green dots and lines*) and shaft synapses (*purple dots and lines*). (D) Plot of normalized uEPSP amplitudes against starting membrane potential for spines (*green dots and lines*) and shaft synapses (*purple dots and lines*).

synapses: 2.01 ± 0.19 spines/10 μm). However, there were several instances (9/23 dendritic segments) in which shaft synapses and spines were observed in equal numbers, or spines outnumbered shaft synapses (data were measured from 23 dendritic segments consisting of total of 1525 μm of dendrite, 317 spines and 392 shaft synapses). Altogether, these data reveal that excitatory synaptic inputs of both structural classes, spines and shaft synapses, coexist and must act together in the integration of synaptic inputs on SNc dopamine neurons.

## Comparison of voltage-dependence of spines and shaft synapses

Dopamine neurons are spontaneously active and therefore, synaptic inputs will be received on an ever changing membrane potential (*Hage and Khaliq, 2015*; *Puopolo et al., 2007*). Therefore, we first examined how steady-state changes in subthreshold potential (−75 to −45 mV) influence

glutamate-evoked signals from spines as compared to shaft synapses (*Figure 8*). Recordings were made in bath applied TTX (500 nM) and nifedipine (10 µM) to limit spontaneous membrane oscillations at depolarized potentials. In Purkinje neurons, T-type $Ca^{2+}$ channels display greater expression within the spines than the dendrites (*Isope et al., 2012*). While the sub-cellular distribution T-type channels has not been examined in dopamine neurons, differential expression in the spines and dendrites could influence the relationship between glutamate-evoked $Ca^{2+}$ influx and membrane potential. Therefore, in a subset of experiments, T-type $Ca^{2+}$ channels were blocked with TTA-P2 (1 µM). Results in the two conditions were similar, and therefore data was pooled (16 spines and 16 shaft synapses recorded with TTA-P2, 15 spines and 10 shaft synapses recorded without TTA-P2). Because of the large spine-to-spine variability in absolute $Ca^{2+}$ signals observed (*Figure 4C*), we normalized the amplitude of glutamate-evoked $Ca^{2+}$ signals to those recorded at −45 mV within the same spine or shaft. This normalization will also account for possible differences in receptor expression between shafts and spines.

In both spines and shaft synapses, glutamate-evoked $Ca^{2+}$ signals increased in amplitude with depolarization, likely due to $Mg^{2+}$ unblock of NMDA receptors (*Figure 8A,B*). Interestingly at hyperpolarized potentials (−60 to −65 mV), the normalized $Ca^{2+}$ signals in spines were significantly larger in amplitude than analogous shaft $Ca^{2+}$ signals (normalized $Ca^{2+}$ signals in spines vs shafts: 0.61 ± 0.03 vs 0.50 ± 0.03; spine, n = 52 measurements, 31 spines; shafts, n = 46 measurements, 21 shafts; p=3.3e-3) (*Figure 8C*). Therefore, spine $Ca^{2+}$ signals were less sensitive to hyperpolarization than shafts signals. We also measured the impact of steady-state voltage changes on the amplitudes of uEPSPs for shaft versus spine synapses (*Figure 8D*). Here, we normalized the amplitudes of all uEPSPs to the amplitude measured at −75 mV, where the driving force of the synaptic current is largest. Between −55 to −50 mV, the average normalized amplitude of shaft uEPSPs was 0.71 ± 0.03 (n = 49 measurements from 26 shafts). By contrast, the average normalized amplitude of spine uEPSPs was 1.06 ± 0.11 (significantly larger than normalized shaft uEPSPs, p=0.004, n = 57 measurements from 31 spines), indicating that there was strikingly little effect of depolarization on the amplitude of spine uEPSPs. This suggests that at hyperpolarized membrane potentials, NMDA receptors are more effectively recruited in spines relative to shaft synapses which we reason may occur through boosting of the voltage in the spine head.

## Glutamate-evoked $Ca^{2+}$ signals in spines during slow tonic firing

Firing in dopamine neurons is shaped by an interaction between synaptic inputs and active subthreshold conductances that drive pacemaking. To better understand the functional contribution of spiny synapses to dopamine neuron excitability, we imaged glutamate-evoked $Ca^{2+}$ responses in spines during slow tonic firing. *Figure 9A* shows AP-evoked $Ca^{2+}$ signals measured during tonic firing for a spine and nearby dendrite. *Figure 9B* shows glutamate uncaging-evoked $Ca^{2+}$ signals and the uEPSP measured while injecting constant negative current to hold the membrane potential near −63 mV. We then uncaged glutamate on a background of tonic firing. We found that when glutamate uncaging shortly preceded an action potential, $Ca^{2+}$ influx into the spine was dramatically larger than the linear sum of the isolated AP- and glutamate-evoked signals (*Figure 9C*). In contrast, when glutamate uncaging occurred shortly after an AP (*Figure 9D left*), spine $Ca^{2+}$ signals were smaller and resembled the predicted linear sum. We measured glutamate-evoked $Ca^{2+}$ signals throughout the entire pacemaker cycle and found that there was a dramatic enhancement of the spine $Ca^{2+}$ signal midway through the firing cycle, much earlier than we had expected (*Figure 9D right*). Notably, the $Ca^{2+}$ signal was enhanced before the onset of the following AP (*open triangle in Figure 9D*), suggesting small changes in subthreshold voltage during tonic firing can dramatically influence synaptic $Ca^{2+}$ influx.

To enable comparison of $Ca^{2+}$ signals across multiple spines, we normalized all uncaging-evoked responses to those recorded from a steady holding potential of -63 mV (*Figure 9E*). Performing this analysis on the linear summations of the isolated AP- and uncaging-evoked $Ca^{2+}$ signals predicts a small phase-dependence of spine $Ca^{2+}$ influx due to additional AP-evoked $Ca^{2+}$ in late phases of the firing cycle (*Figure 9E*). Values slightly <1 are observed early in the firing cycle due to the decay of the AP-evoked $Ca^{2+}$ transient. We observe a much more dramatic influence of phase on glutamate-evoked spine $Ca^{2+}$ signals within our measured data (*Figure 9E*). When glutamate uncaging occurred in the last third of the spike cycle, spine $Ca^{2+}$ influx was 1.90 ± 0.07 fold larger than $Ca^{2+}$ influx produced by uncaging alone. By contrast, linear summation of the AP-evoked and uncaging

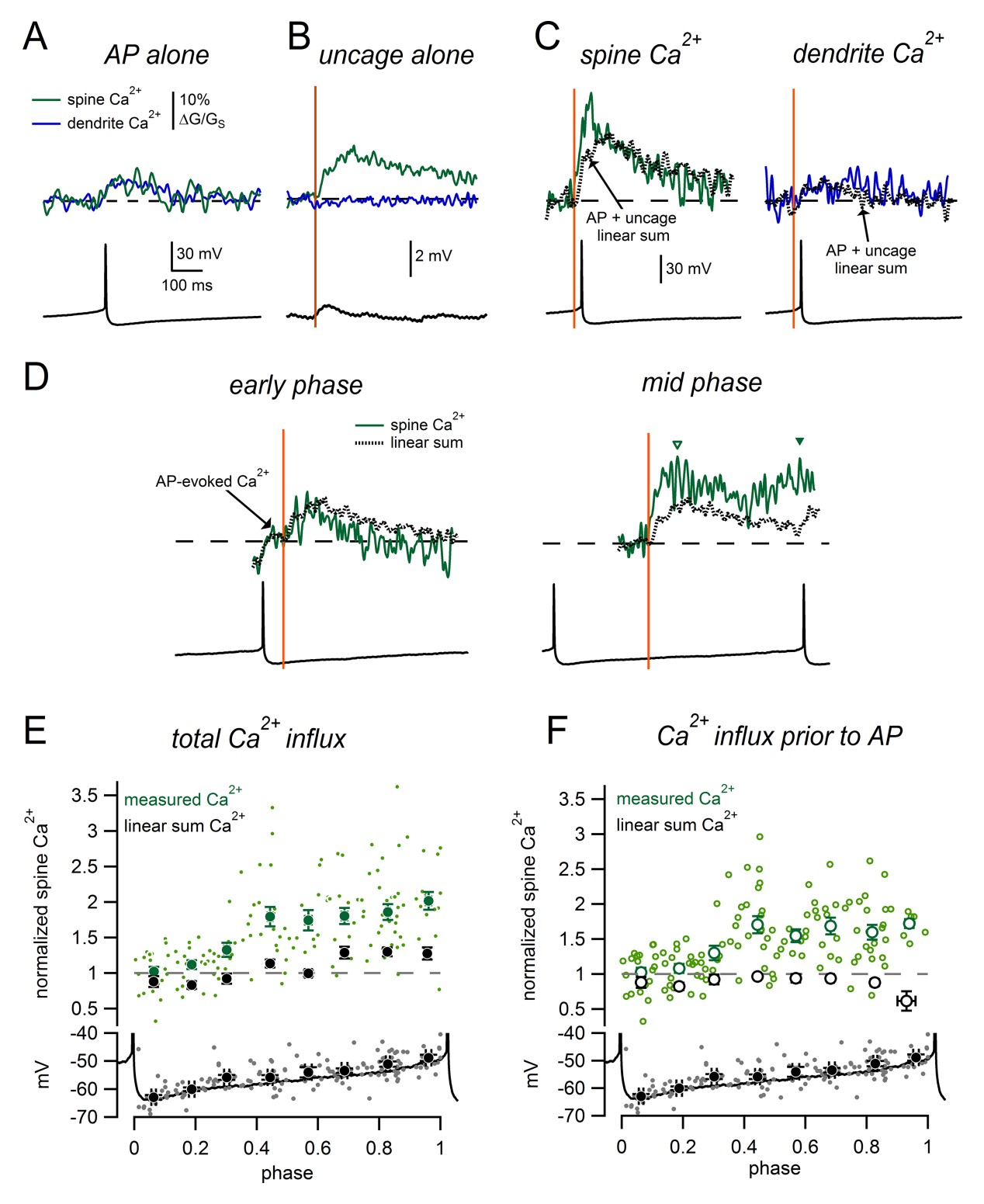

**Figure 9.** Phase-dependent enhancement of glutamate-evoked Ca²⁺ signals during tonic firing. (A) AP-evoked Ca²⁺ signals measured in a spine (*green*) and nearby dendrite (*blue*) during tonic firing. Signals are averages of 3 individual AP-evoked Ca²⁺ transients. (B) Ca²⁺ signals as in A, for glutamate uncaging near the spine head while holding at −63 mV. Signals are averages of 6 uncaging trials. (C) Spine and dendrite Ca²⁺ signals evoked by uncaging glutamate during tonic firing. Dotted lines represent the 'linear sum', defined as 'AP-alone' plus 'uncage- alone' signals from panels A and B. (D) Comparison of spine Ca²⁺ signals and corresponding linear sums for trials in which glutamate uncaging occurred at either an early phase (*left*) or intermediate phase (*right*) of the firing cycle. Data are from the same spine as panels A–C. Dashed lines indicate baseline signals before the uncaging

*Figure 9 continued on next page*

*Figure 9 continued*

pulse. In the *early phase* example, the Ca$^{2+}$ signal displayed an AP-evoked increase before the uncaging pulse that was not measured as part of the glutamate-evoked response (arrow). Inverted triangles in the *mid-phase* example indicate the peaks of the glutamate-evoked Ca$^{2+}$ signal measured before (*open symbol*) and after the subsequent AP (*closed symbol*). (E) *Top*: Normalized spine Ca$^{2+}$ amplitude plotted against the phase at which glutamate uncaging occurred for measured data (*green*) and linear sums (*black*). *Bottom*: Plot of membrane potential immediately prior to glutamate uncaging versus phase. The black trace shows a typical interspike interval. (F) As in E, except glutamate-evoked Ca$^{2+}$ signals were only measured before the onset of the subsequent AP.

evoked signals during this same range of the spike cycle predicts signals just $1.28 \pm 0.04$ fold larger than uncaging alone (measured data was significantly greater than linear sums, p=1.4e-9, n = 58 measurements from 20 spines). Action potentials immediately following synaptic activation are known to enhance Ca$^{2+}$ signals (*Nevian and Sakmann, 2004*). However, we wanted to determine if the subthreshold voltage alone was sufficient to enhance glutamate-evoked Ca$^{2+}$ influx. To isolate the effect of subthreshold voltage on spine Ca$^{2+}$ signals, we measured the peak Ca$^{2+}$ signals before the onset of spikes and excluded Ca$^{2+}$ signals if spikes occurred <100 ms after the uncaging pulse (*Figure 9F*). Using this alternative measure, we observed dramatic enhancement of spine Ca$^{2+}$ influx in the middle and late phases of the spike cycle (middle phase: measured Ca$^{2+}$ = $1.67 \pm 0.07$, linear sum = $0.95 \pm 0.03$, p=5.5e-12, n = 41 measurements from 20 spines; late phase: measured Ca$^{2+}$ = $1.64 \pm 0.07$, linear sum = $0.85 \pm 0.04$, p=2.7e-13, n = 36 measurements from 20 spines).Therefore,

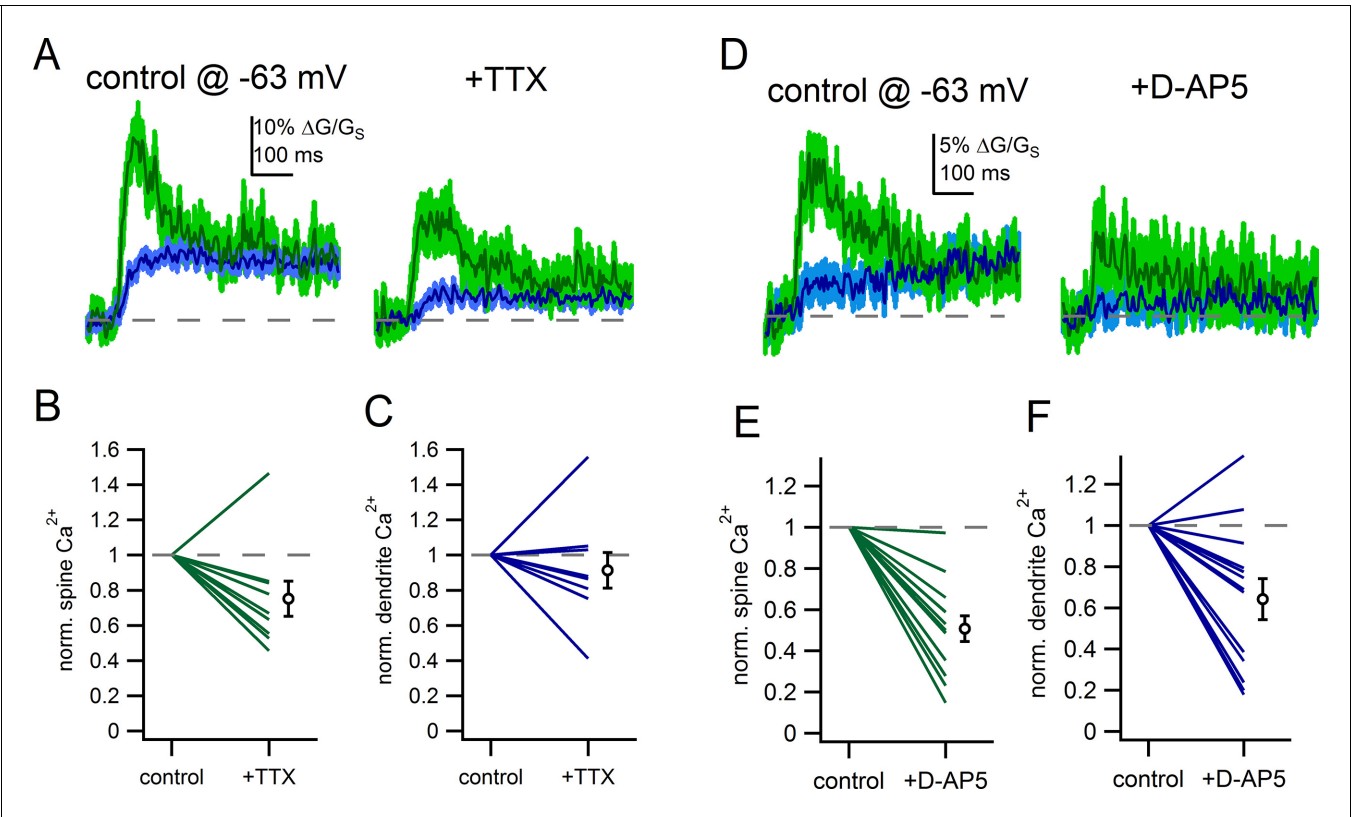

**Figure 10.** Voltage-gated sodium channels and NMDAR shape spine Ca$^{2+}$ signals. (A) Example of uncaging evoked Ca$^{2+}$ signals in control conditions and following application of TTX (500 nM). (B) Plot of the effect of TTX on spine Ca$^{2+}$ signals. (C) Plot of the effect of TTX on dendrite Ca$^{2+}$ signals. (D) Example of uncaging evoked Ca$^{2+}$ signals in control conditions and following application of D-AP5 (50 µM). (E) Plot of the effect of D-AP5 on spine Ca$^{2+}$ signals. (F) Plot of the effect of D-AP5 on dendrite Ca$^{2+}$ signals.

we find that glutamate-evoked spine $Ca^{2+}$ influx in dopamine neurons is enhanced in a time window that occurs periodically during the natural pacemaker cycle.

## Voltage-gated sodium channels and NMDA receptors shape spine $Ca^{2+}$ signals

Finally, if the spine neck resistance results in amplification of EPSPs within the spine head (*Figure 8*), then voltage-gated channels may be more likely to be activated and shape the synaptic response. To determine whether voltage-gated sodiumchannels contribute to synaptic responses of spines, we measured the effect of TTX (500 nM) on glutamate-evoked spine $Ca^{2+}$ signals from a hyperpolarized membrane potential (-63 mV). In the example spine show in *Figure 10A*, application of TTX reduced the amplitude of the spine $Ca^{2+}$ signal from 30% $\Delta G/G_S$ to 17% $\Delta G/G_S$. On average, TTX reduced the spine $Ca^{2+}$ signal by $25 \pm 10.0\%$ (p=0.039, student's paired) (*Figure 10B*) but had little effect on dendritic $Ca^{2+}$ signals (reduced by $9 \pm 10\%$, p=0.45) (*Figure 10C*). Therefore, in addition to the known involvement of sodium channels in enabling robust backpropagation of action potentials in the dendrites of dopamine neurons (*Gentet and Williams, 2007*; *Hausser et al., 1995*), we find a likely role of sodium channels in boosting synaptic potentials in spines of substantia nigra dopamine neurons.

In separate experiments, we tested the effect of blockade of NMDA receptors with D-AP5 (50 µM). In the example shown in *Figure 10D*, D-AP5 reduced the glutamate-evoked spine $Ca^{2+}$ signal from 12% $\Delta G/G_S$ to 6% $\Delta G/G_S$. On average, blockade of NMDA receptors reduced spine $Ca^{2+}$ influx by $49 \pm 6\%$ (p=2.2e-3) (*Figure 10E*). The effect of D-AP5 on the dendrite $Ca^{2+}$ signal was smaller, on average reduced by $36 \pm 10\%$ (p=0.02) (*Figure 10F*). The finding that TTX and D-AP5 produced greater attenuation of the spine $Ca^{2+}$ signal than the dendritic $Ca^{2+}$ signal suggests that there is greater activation of NMDA receptors and voltage-dependent channels in the spine head than the dendrite in response to glutamate uncaging. The contribution of voltage-gated channels to glutamate-evoked responses provides a mechanism by which subtle changes in the subthreshold voltage—like those observed during tonic firing—can shape the synaptic responses of dendritic spines.

## Discussion

We show that dendritic spines are present on substantia nigra dopamine neurons both in live and in fast-perfusion fixed tissue. In PSD-95 mVenus mice, spines were present at densities only slightly lower than shaft synapses. Importantly, electrical stimulation of presynaptic inputs resulted in $Ca^{2+}$ influx into the spine head demonstrating that spines are innervated by terminals and act as functional sites of synaptic transmission. Testing individual spines with glutamate uncaging, we show that activation of long-necked spines results in small amplitude uEPSPs due in part to a lower AMPA/NMDA ratio and higher spine neck resistance. We demonstrate that stronger electrical compartmentalization in spines results in boosting of the spine voltage. Lastly, activating spines during tonic firing, we found that spine $Ca^{2+}$ signals are dramatically enhanced during the middle phase of the interspike interval, independent of action potentials. This enhancement of spine $Ca^{2+}$ results from strong recruitment of NMDA receptors and voltage-gated channels, likely due to boosting of the spine voltage. Therefore, these results show that in addition to shaft synapses, a meaningful source of excitation of SNc arrives via synaptic inputs onto dendritic spines.

### Density of dendritic spines on substantia nigra dopamine neurons

Our results are in accord with a number of past studies that have observed dendritic spines on midbrain dopamine neurons. To date, spines have been identified on dopamine neurons in rat (*Grace and Onn, 1989*; *Nirenberg et al., 1996b*; *Phillipson, 1979*; *Sarti et al., 2007*), guinea pig (*Yung et al., 1991*), cat (*Phelps et al., 1983*; *Preston et al., 1981*; *Rinvik and Grofova, 1970*), rabbit (*Kline and Felten, 1985*), primates (*Schwyn and Fox, 1974*) and humans (*Cruz-Sanchez et al., 1995*; *Patt et al., 1991*). We estimated spine densities from dendritic segments and whole-cell reconstructions and found that dendritic spines are present at an average density of 2.08 spines/10 µm. These values are also in close agreement with a recent study of dopamine neurons spines in juvenile mice (*Jang et al., 2015*). Along with past work, therefore, our data provide clear evidence that synaptic excitation of dopamine neurons likely involves shaft synapses as well as input onto dendritic spines.

We examined dopamine neurons in both live slices as well as fast-perfusion fixed tissue and found a modest but consistent reduction in spine density with age. This finding is consistent with developmental studies that report fewer spines on dopamine neurons in adult animals (*Phelps and Adinolfi, 1982*; *Phelps et al., 1983*). Humans, however, may be a notable exception to this rule according to two studies that characterized the morphology of dopamine neurons in substantia nigra from adult deceased patients (*Cruz-Sanchez et al., 1995*; *Patt et al., 1991*). Both studies report significant numbers of dendritic spines in normal, nondiseased adult patients (age range, 20–93 years old). Interestingly, one of these studies reported that the density of spines on putative SNc dopamine neurons in normal adults was 1–2 spines/10 µm (*Cruz-Sanchez et al., 1995*), in striking agreement with the values that we report here in mice. In another study, the authors observed prominent pathological changes to dopamine neurons in Parkinson's patients including reduction in dendritic branching and a loss in the number of dendritic spines (*Patt et al., 1991*). In future work, it will be important to determine whether a causative relationship exists between the observed changes in dendritic morphology/spine density and selective death of substantia nigra dopamine neurons which occurs in Parkinson's patients.

Comparing dopamine neurons to other well-studied spiny neurons reveals similarities but also important differences. For example, pyramidal neurons receive the vast majority (>85%) of excitatory inputs onto spines (*Kasthuri et al., 2015*) while dopamine neurons are unusual in that they receive inputs onto a mixture of shaft and spine synapses. The density of spines on dopamine neurons is considerably lower than pyramidal neurons (typically >10 spines/10 µm) but similar to some GABAergic interneurons neurons of the hippocampus and cortex that have spine densities of 2–4 spines/10 µm (*Kawaguchi et al., 2006*; *Scheuss and Bonhoeffer, 2014*). It also must be noted that unlike cortical and hippocampal neurons that receive mostly excitatory input, substantia nigra dopamine neurons function within a largely inhibitory network. Up to 70% of synaptic inputs that are received by the SNc dopamine neurons are inhibitory while a much smaller fraction of the synaptic inputs arrive from excitatory sources (*Bolam and Smith, 1991*; *Henny et al., 2012*; *Smith et al., 1996*). This raises the possibility that the low density of dendritic spines on SNc dopamine neurons may simply reflect the low density of excitatory inputs overall. Consistent with this idea, we find that the ratio of spines to PSD95-mVenus positive shaft puncta is 4 to 5. Therefore, despite a low absolute density of spines on dopamine neurons, the relative density of shaft and spine synapses is comparable.

## Spine morphology and electrical compartmentalization

Whether long-necked spines more strongly attenuate synaptic potentials is a matter of current debate. Studies of cortical pyramidal cells show a negative correlation between neck length and EPSP amplitude (*Araya et al., 2006*; *2014*), while studies in hippocampal pyramidal cells (*Takasaki and Sabatini, 2014*) show only a weak correlation. It is important to note that Takasaki and Sabatini tested spines that were below 1.2 µm in length whereas Araya et al. tested substantially longer spines ($\geq$2 µm). In the dopamine neurons, we found a wide range of spine lengths up to 5 µm. However if we only consider spines shorter than 1.2 µm, we also observe no significant correlation between spine length and uEPSP amplitude (Pearson's R = −0.054; n = 21 spines; p=0.81). In a different study, *Bywalez et al. (2015)* found no correlation between spine length and EPSP amplitude in olfactory bulb granule neurons, which have spines up to 15 µm in length. However, spines on the granule cells are excitable and possess voltage-gated sodium and calcium channels which shape the release of neurotransmitter from the spines (*Egger et al., 2005*). As Bywalez et al. proposed, the spine sodium channels may function to reduce variability in spine potentials, $Ca^{2+}$ influx and subsequent neurotransmitter release from spines.

## Tonic firing allows dopamine neurons to capitalize on spine neck amplification

The dendrites of many neurons are excitable and express a wide variety of voltage-gated ion channels that shape action potential backpropagation and synaptic integration (*Stuart and Spruston, 2015*). However, there is not a consensus regarding whether or not stimulation of a single spine leads to downstream activation of voltage-gated ion channels. Studies using voltage-sensitive fluorescent indicators to assay spine voltage report that the depolarization of the spine head following single spine stimulation is insufficient for activation of voltage-gated channels (*Palmer and Stuart,*

*2009*; *Popovic et al., 2015*). By contrast, $Ca^{2+}$ imaging studies examining single spine responses have described a clear contribution of voltage-gated ion channels in the spine head with many observing activation of high-voltage activated calcium channels (*Bloodgood et al., 2009*; *Bywalez et al., 2015*; *Carter and Sabatini, 2004*; *Grunditz et al., 2008*; *Harnett et al., 2012*; *Seong et al., 2014*). These observations predict that the electrical resistance of the spine neck amplifies the EPSP in the spine head to potentials considerably greater than those recorded in the dendrite or soma.

We observed a significant contribution of voltage-gated sodium channels to the glutamate-evoked responses of individual spines (*Figure 10*), suggesting the depolarization of the spine head is sufficient for sodium channel activation in dopamine neurons. Most studies of spine function have examined pyramidal neurons that rest at hyperpolarized membrane potentials near -70 mV. However, during pacemaking in dopamine neurons, the interspike membrane potential covers a narrow but relatively depolarized subthreshold voltage range with an average non-spike voltage of ∼-55 mV (*Hage and Khaliq, 2015*). At these depolarized voltages, the membrane potential is within the steepest region of the voltage-activation curve of sodium channels, where even a small amplification of the spine head EPSP may lead to significant sodium channel activation. In a clear demonstration of the relationship of EPSPs with voltage-gated sodium channels, *Carter et al. (2012)* showed that EPSPs of 5 mV can lead to activation of both persistent and transient sodium current. Therefore, even if the effects of the spine neck resistance on electrical signaling are small, the intrinsic properties of dopamine neurons may allow minor amplification to produce major effects on the synaptic voltage response.

## Materials and methods

### Slice preparation

For experiments utilizing wildtype animals, sagittal brain slices containing SNc were prepared from postnatal day 14–25 Swiss Webster mice of either sex according to the institutional guidelines at the National Institutes of Health. Experiments utilizing the PSD-95-ENABLED mouse line performed on juvenile mouse progeny generated by crossing heterozygous PSD-95-ENABLED mouse with a heterozygous DAT-Cre mouse. Mice were genotyped using previously published primers (*Fortin et al., 2014*). Animals were anesthetized with isoflurane and swiftly decapitated. Brains were quickly removed and placed in ice-cold slicing solution containing (in mM) 250 glycerol, 2.5 KCl, 2 $MgCl_2$, 2 $CaCl_2$, 1.2 $NaH_2PO_4$, 10 HEPES, 21 NaHCO3, 5 glucose, bubbled with 95/5% $O_2/CO_2$. Slices were cut at 300 µm thickness using a vibrotome (DTK-1000; DSK, Dosaka) and incubated for 30 min at 34° C in artificial cerebrospinal fluid (ACSF) containing (in mM) 125 NaCl, 25 $NaHCO_3$, 1.25 $NaH_2PO_4$, 3.5 KCl, 1 $MgCl_2$, 2 $CaCl_2$, and 10 glucose, bubbled with 95/5% $O_2/CO_2$. Slices were then stored at room temperature until time of use.

### Electrophysiological recording

Slices were placed into a recording chamber and perfused continuously with warm ACSF (32–34°C). Dopamine neurons were targeted primarily by their location within the SNc. Other criteria included the presence of slow pacemaking (<5 Hz) during cell-attached or whole-cell recordings, broad APs (halfwidth >1.35 ms) and prominent voltage sag in response to negative current injection—associated with hyperpolarization-activated cation current ($I_H$). Current-clamp and voltage-clamp recordings were made with a Multiclamp 700B amplifier and digitized using a Digidata 1440A (Molecular Devices). Low-resistance patch electrodes (2–4 MΩ) were pulled from filamented borosilicate glass. In voltage-clamp recordings, pipette series resistance was compensated by 70–80% and was monitored carefully throughout the experiment. Recordings were terminated if series resistance changed by >20%. Current-clamp recordings were bridge balanced and monitored frequently throughout the experiment. For current-clamp recordings, internal recording solution contained (in mM) 122 K methanesulfonate, 9 NaCl, 1.8 $MgCl_2$, 4 Mg-ATP, 0.3 Na-GTP, 14 phosphocreatine, 10 HEPES, 0.5 EGTA, 0.1 $CaCl_2$ and 0.05 Alexa Fluor 594 hydrazide (Molecular Probes, Eugene, OR) adjusted to 7.35 with NaOH. For $Ca^{2+}$ imaging experiments, EGTA and $CaCl_2$ were excluded from the internal solution and 300 µM Fluo-5F ($K_D$ = ∼ 2.3 µM) was added. For voltage-clamp recordings internal solution contained (in mM) 135 CsCl, 10 NaCl, 10 HEPES, 2 $MgCl_2$, 0.5 EGTA and 0.1 $CaCl_2$, except

where specifically indicated. In some cases, recordings were made using QX-314 (1 mM) added to the internal solution as well as bath applied nifedipine (10 µM) to reduce spontaneous membrane oscillations when holding at depolarized potentials (*Puopolo et al., 2007*).

## Two-photon laser scanning microscopy and glutamate uncaging

Imaging experiments were performed using a custom two-photon microscope from Prairie Technologies Ultima (Middleton WI) along with a Mai Tai ultrafast Ti:Sapphire laser (Spectra-Physics, Mountain View CA) tuned to 810 nm for $Ca^{2+}$ imaging. For experiments using the PSD-95-ENABLED mice the excitation laser was tuned to 960 nm to visualize the mVenus signal. Cells were imaged using a 40x, 0.8 NA objective (Olympus, Melville NY). Fluorescence was split into red and green channels using a 575 nm dichroic longpass mirror and passed through 607/45 nm and 525/70 nm barrier filters before being detected by multi-alkali photomultiplier tubes (Hamamatsu). $Ca^{2+}$ imaging was initiated 15–20 min after whole-cell break in to allow diffusion of fluorescent indicators. Linescan imaging of dendritic $Ca^{2+}$ was performed at 30 s intervals. $Ca^{2+}$ imaging data are presented as $\Delta G/G_S$ and were quantified as changes in green fluorescence divided by red fluorescence ($\Delta G/R$), normalized to $G_S/R * 100\%$. $G_S/R$ was measured by imaging a pipette filled with internal recording solution plus saturating $Ca^{2+}$ (2 mM $CaCl_2$) placed directly above the slice at the end of experiment (*Yasuda et al., 2004*).

Simultaneous glutamate uncaging experiments were performed using a second Mai Tai laser tuned to 725 nm with an uncaging pulse width of 500 µs. Uncaging laser power measured at the back of the objective was between 35 and 45 mW. Synapses assayed by glutamate uncaging were 30–40 µm below the surface of the slice. The same parameters were used to bleach the spine head in FRAP recordings. The extent of bleaching by the uncaging laser was consistent from spine to spine (fluorescence reduced by 34 ± 2%), suggesting effective uncaging power was similar across experiments. External solutions for glutamate uncaging experiments contained 3 mM 4-Methoxy-7-nitroindolinyl-caged-L-glutamate (MNI-glutamate) (Tocris bioscience) and 10 µM D-serine to prevent NMDAR desensitization. Solutions were recirculated to conserve MNI-glutamate (volume = 10 mL).

## Local extracellular stimulation

Local stimulation of dendritically-located synapses was performed using bipolar electrodes placed in theta glass pipettes filled with ACSF (tip diameter ~ 5 µm). Electrodes were placed within 10 µm of an Alexa-594 labeled dendrite. Stimulus intensity was set to 15–45 V with 0.5 ms pulse duration using an Iso-Flex stimulus isolator (A.M.P.I.). Picrotoxin (50 µM), CGP 55,845 hydrochloride (1 µM), sulpiride (1 µM), SCH 39,166 hydrobromide (1 µM) and LY 341495 (1 µM) were added to ACSF to block activity of $GABA_A$, $GABA_B$, $D_2$ dopamine, $D_1$ dopamine and group II metabotropic glutamate receptors.

## Morphological examination and analyses

Golgi-Cox staining of Swiss Webster mouse brain tissue was performed using FD Rapid GolgiStain Kit (FD NeuroTechnologies, Inc., Columbia MD) according to manufacturer instructions. Juxtacellular labelling of SNc dopamine neurons was performed using Tg(TH-EGFP)1Gsat GENSAT line mice which express enhanced GFP under the control of the tyrosine hydroxylase promoter. Mice were transcardially perfused with cold 1.5% PFA in phosphate buffered saline (PBS) after which brains were dissected and postfixed for 1 hr in 1.5% PFA. Brains were sectioned at 300 µm. Fixed brain slices containing the SNc were transferred to a microscope stage and dopaminergic neurons were identified by green fluorescence. A high resistance sharp electrode containing 0.5 mM Alexa-594 was then placed against the soma of an identified dopamine neuron and large, brief (500 µs) current pulses were applied once every 30 s for 5–10 min until the soma was clearly filled with Alexa-594. Afterwards, the slice was mounted on a glass slide with VectaShield mounting medium (Vecta) and imaged on a two photon microscope (described above).

Images used for measurements of spine length and density along dendritic segments were performed using z-stacks of 64 $nm^2$xy-resolution. Images used to measure spine distribution along entire dendrites had a resolution of 146 $nm^2$. Planes of z-stacks were separated by either 0.8 or 1.0 µm. When dendritic regions examined extended beyond the borders of a single z-stack, images were integrated using the Volume Integration and Alignment System (VIAS) software package

(*Rodriguez et al., 2003*). The combined data set was then loaded into Neuron Studio (*Wearne et al., 2005*) where dendrites were semi-automatically reconstructed and dendritic spines were manually identified.

Spine morphology was measured in ImageJ. The spine length was measured from the base of the dendrite to the tip of the spine head. Spine head diameter was estimated by measuring full-width at half maximum of the pixel intensity across a 3 pixel wide line drawn orthogonal to the spine neck. The spine head diameter was then used to estimate the spine head volume by $V_{head} = 4/3*\pi*r^3$, where r is the radius of the spine head.

The pixel intensity of mVenus signal in PSD-95-ENABLED mice was measured by drawing an ROI around the dendritic shaft or spine of interest and measuring the pixel intensity throughout the z-stack. After background subtraction, green and red pixel intensities were measured as the integral of 3 consecutive z-slices, with the middle slice corresponding to the highest red intensity. Bleed-through of Alexa-594 signal into the green channel was measured in regions of the dendritic shaft with no apparent mVenus signal (average bleedthrough = 3.3%). mVenus pixel intensities for dendritic shafts and spines were therefore corrected by subtracting the product of the corresponding red pixel intensity and the bleedthrough measured for that dendrite.

Electrophysiological data were analyzed using custom routines written in Igor Pro (Wavemetrics). $Ca^{2+}$ imaging and FRAP data were quantified using ImageJ to measure fluorescence intensities and further analyzed using Igor.

FRAP data were corrected for slight and gradual acquisition- related bleaching due to scanning by the imaging laser (*Ishikawa-Ankerhold et al., 2012*). To isolate the extent of acquisition-related bleaching, we measured the Alexa-594 fluorescence intensity during linescans in which the uncaging laser was not targeted to the spine head. On average, the baseline fluorescence was reduced by 3.5 ± 0.7% (n=33) over the course of imaging. These acquisition-only signals were fit with a straight line and FRAP traces were normalized to these functions. Corrected FRAP traces were then fit with a double-exponential function constrained to return to baseline fluorescence levels. The weighted time constant was calculated as:

$$\frac{A_{fast} \times \tau_{fast} + A_{slow} \times \tau_{slow}}{A_{fast} + A_{slow}}.$$

## Acknowledgement

We thank Rebekah Evans and Rahilla Tarfa and other members of the Khaliq laboratory for helpful comments on the manuscript. We also thank Garland Kennedy for his assistance in obtaining spine density counts during an earlier phase of this project. We thank Dr. Haining Zhong for providing *PSD-95-ENABLED* mice.

## Additional information

### Funding

| Funder | Grant reference number | Author |
| --- | --- | --- |
| National Institutes of Health | NS003135 | Travis A Hage<br>Yujie Sun<br>Zayd M Khaliq |

The funders had no role in study design, data collection and interpretation, or the decision to submit the work for publication.

### Author contributions

TAH, YS, Conception and design, Acquisition of data, Analysis and interpretation of data, Drafting or revising the article; ZMK, Conception and design, Analysis and interpretation of data, Drafting or revising the article

### Author ORCIDs

Zayd M Khaliq, http://orcid.org/0000-0002-1445-1457

## Ethics

Animal experimentation: This study was performed in strict accordance with the institutional guidelines recommended by the National Institutes of Health. All animals used were handled according to institutional animal care and use committee (IACUC) protocol ASP#1332 of NIH/NINDS.

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
