## [Decision Letter]

Thank you for submitting your work entitled "Electrical and Ca^2+^ signaling in dendritic spines of substantia nigra dopaminergic neurons" for consideration by *eLife*. Your article has been reviewed by two peer reviewers, and the evaluation has been overseen by Gary Westbrook as the Senior Editor and Reviewing Editor. The reviewers have discussed the reviews with one another and the Reviewing Editor has drafted this decision to help you prepare a revised submission.

Summary:

This manuscript examines an interesting and somewhat understudied topic – the role of dendritic spines on dopamine neurons, which are outnumbered by shaft synapses by about 4-fold. Both reviewers were supportive and ask for revisions that should be addressable primarily with analysis of existing data and a more thorough description of methods. The comments to be addressed are included below. Some of reviewer 2's comments may require some additional experiments (e.g. points 2,4,5,7) if the data are not available.

*Reviewer #1:*

1) This is a comprehensive examination of the sites involved in glutamate dependent transmission on dopamine neurons. Functional synapses on the spines dendritic shafts of dopamine neurons have not been compared. The differences between the two types of synapses is novel and important. The results of this study may explain multiple observations surrounding the expression of different forms of synaptic plasticity on dopamine neurons.

2) The section on electrical and chemical compartmentalization needs an explanation of the significance of these experiments linked to the functional difference between spine and shaft synapses. It seems obvious but without a better discussion of this issue it is just a lot of work to confirm what has been reported in other places.

3) Figure 9 is a bit confusing when considering the increase in calcium that results from the action potentials. It would be less confusing if the action potential dependent rise in calcium was also illustrated. Presumably that is the point of the open triangle in Part B? It would be helpful to see it in parts A and C. Previous work in this lab reported similar calcium boosting and an increase in transmission in dendrites during pacemaking. The significance of the present results is that the calcium boosting is statistically larger in the spines (compare F&G with I&J).

Reviewer #2:

Hage and colleagues use a combination of electrophysiology and 2photon glutamate uncaging/Ca imaging to study the role of spines on substantia nigra dopaminergic neurons. Evaluating the functional role of spines on various neuron subtypes, including sparsely spiny neurons, is of general importance, will deepen our understanding of spine biology and essential for understanding the specific synaptic computations performed by a given neuronal subtype. That said, there are a few significant issues with the execution of the experiments, the analysis, and interpretation of the results.

1) The authors standardize laser power at the back focal plane of the objective but laser power delivery to the spine/dendrite will decrease significantly with depth into the slice. Experiments should be done either within a relatively tight range of depths or effective laser power should be normalized at the depth of the spine. Overall the power delivery should be more clearly described with respect to the depth of the spine in the slice.

2) Spines with long necks are described as having immature (weak) synapses and high resistance necks that would filter EPSPs. It's not possible to attribute the small EPSPs recorded at the soma to spine neck resistance without knowing if there are AMPARs in the spine head and, if there are AMPARs, how the receptor number compares to that of shorter neck spines. Experiments should be conducted to measure/estimate AMPAR and NMDAR numbers for spines of varying lengths. Fluor intensity of PSD-95-venus puncta might be a suitable proxy.

3) FRAP analysis needs to be revisited with fits constrained to baseline fluorescence. The authors use FRAP measurements (proxy for chemical compartmentalization) to estimate spine neck resistance. In the FRAP example shown, 15-20% of the fluorescence signal does not recover, yet the authors fit this trace with a monoexponential function, that is not constrained to baseline fluorescence and report a tau of 30 ms. (This trace is better described with a double exponential, with the first time constant possibly reflecting diffusion of fluorophore within the spine and the second reflecting diffusion between the spine and dendrite.). If the fluorescence of an untethered molecule doesn't recover, there is a stronger barrier to diffusion than reported. How many of the data points are affected by this? Additionally, spine neck resistance is proportional to the length *cross sectional area* axial resistance. The spine neck diameter is going to have a much larger impact on resistance than length. Finally, the relationship that is most relevant, and least subject to assumptions based on unmeasurable physical features (at least with 2P) would be the FRAP versus uEPSP amplitude. A better experiment altogether would be to uncage (in a standardized way) on either side of the neck and measure a voltage dependent process in the spine head. This has been done by the Sabatini and Magee labs to good effect.

4) The authors report there is an increase AMPA to NMDA ratio in shorter spines. The data presented in 6C indicate that on average the ratio is ~ equivalent. Moreover, the authors need to distinguish between 1) differences in receptor number at synapses on long versus short spines, 2) differential filtering of fast AMPA versus slow NMDAR mediated potentials, and 3) over all compartmentalization of electrical and biochemical signals within the spine, via the spine neck. Specifically, are AMPA and NMDA electrical signals equally attenuated by the spine neck?

5) Experiments testing the ohmic nature of spine neck resistance should be shown as well as described in text.

6) If PSD-95-venus fluorescence correlates with uEPSP amplitude (and presumably AMPAR number) how does venus intensity correlate with spine length? This may provide a strong paradigm for disambiguating the issues in points (3) and (4) The authors might have these experiments done already but only need to analyze them. If the long spines have no PSD-95 puncta, they might not have AMPARs, so stating the spine neck filters these currents is inaccurate.

7) The authors uncage on the spine and dendrite while the cell is at different membrane potentials. They observe the spine Ca signal is larger at more negative membrane potentials and attribute this to increase relief of Mg block at spine synapses due to the larger depolarization of the spine head. Although this is a possible interpretation, it is also possible that there are T-type VGCCs in the spine head that are not present in the adjacent shaft. Experiments should be done to determine if the spine and dendrite have the same NMDAR and VGCC mediated Ca signals. If the Ca sources are different the measurements they've made might not be correctly interpreted.

8) What does the Ca signal in the spine and dendrite look like during tonic firing without uncaging? The authors should show Ca transients from spikes alone.

---

## [Author Response]

*Reviewer #1:*

1) This is a comprehensive examination of the sites involved in glutamate dependent transmission on dopamine neurons. Functional synapses on the spines dendritic shafts of dopamine neurons have not been compared. The differences between the two types of synapses is novel and important. The results of this study may explain multiple observations surrounding the expression of different forms of synaptic plasticity on dopamine neurons.

We appreciate the positive comments provided by the reviewer.

2) The section on electrical and chemical compartmentalization needs an explanation of the significance of these experiments linked to the functional difference between spine and shaft synapses. It seems obvious but without a better discussion of this issue it is just a lot of work to confirm what has been reported in other places.

We have added a brief discussion to the relevant section of the Results (copied below).

“The presence of glutamatergic synapses on both dendritic shafts and spines in dopamine neurons raises the possibility that these two structural classes of synapses could differ in chemical and electrical signaling. […] Finally, the spine neck could produce local boosting of the EPSP within the spine head leading to activation of voltage-dependent ion channels that would not take place at shaft synapses.”

3) Figure 9 is a bit confusing when considering the increase in calcium that results from the action potentials. It would be less confusing if the action potential dependent rise in calcium was also illustrated. Presumably that is the point of the open triangle in Part B? It would be helpful to see it in parts A and C. Previous work in this lab reported similar calcium boosting and an increase in transmission in dendrites during pacemaking. The significance of the present results is that the calcium boosting is statistically (?) larger in the spines (compare F&G with I&J).

We agree with the viewpoint presented here and thank the reviewer for the opportunity to clarify this issue. We now illustrate isolated AP-evoked and uncaging-evoked Ca^2+^ signals measured in the spine and dendrites (Figure 9). We have conducted further analyses in which we made linear predictions of spine Ca^2+^ signals during pacemaking by summing the two isolated signals according to the timing of glutamate uncaging and action potential firing. These “linear sums” are now shown with the recorded spine Ca^2+^ signals for measurements made during 3 different phases of pacemaking (Figure 9). Examining the relationship between the expected linear sums and the phase of the spike cycle predicted a small enhancement of spine Ca^2+^ influx in late phases due to the additional AP-evoked Ca^2+^ influx (Figure 9). The measured enhancement of Ca^2+^ influx was significantly larger (Figure 9) and could be observed even if when we limited our analyses to Ca^2+^ influx occurring before the AP (Figure 9). We indicated in the figure legend that the open and closed triangles are illustrating peak Ca^2+^ signals measured before and after the AP.

We have made Figure 9 examining the effect of TTX and DAP5 on glutamate-evoked Ca^2+^ influx into the spine and dendrite a separate figure (Figure 10). These experiments were not performed in tonically active cells. We originally included these data with experiments examining pacemaking as we believe VGSCs and NMDARs contribute to the observed phase-dependence of spine Ca^2+^ influx as described in the Results and Discussion.

We agree that the findings shown here share some important similarities and differences to the findings reported in Hage and Khaliq, 2015. The previous study reported that increases in the rate of tonic firing could influence retrograde Ca^2+^ signaling and synaptically-evoked bursts. This was due to a fairly small, but fairly steady depolarization of the average interspike membrane potential. This depolarization promoted activation of L-type Ca^2+^ channels and NMDARs.

In this study, we examined the influence of tonic firing on synaptic Ca^2+^ signaling in individual spines and on a much finer time scale. Similar to the changes observed with long-lasting increases in tonic firing rate reported in Hage and Khaliq 2015, the enhancement of spine Ca^2+^ at late phases of the spike cycle that we observe here is due to a more depolarized membrane potential – promoting activation of NMDARs and voltage-gated channels. While our previous study predicted a possible phase dependent Ca^2+^ enhancement also through depolarization, we were surprised both by the magnitude and timing of the enhancement in the present experiments. In addition, we believe that a key contributor to the enhancement of spine Ca^2+^ in this study is passive amplification that occurs as a result of the spine neck resistance.

Reviewer #2:

*1) The authors standardize laser power at the back focal plane of the objective but laser power delivery to the spine/dendrite will decrease significantly with depth into the slice. Experiments should be done either within a relatively tight range of depths or effective laser power should be normalized at the depth of the spine. Overall the power delivery should be more clearly described with respect to the depth of the spine in the slice.*

The reviewer raises an important methodological point. Throughout this study, we targeted spines 30-40 µm below the surface of the slice and have now included this statement in the methods. Consistency in laser power delivered to the spine was confirmed in FRAP experiments. For these experiments, we used the same uncaging protocol that was used to measure the EPSP and spine Ca^2+^ influx. The extent of bleaching for a 0.5 ms pulse (35-45 mW at the back aperture) was largely consistent at 34 ± 2% (n=33 spines).

The issue raised by the reviewer also highlights the importance of our paired comparisons of short versus long spines, or spines versus shaft synapses. Neighboring postsynaptic structures at the same depth in slice and the same distance from the soma were assayed using identical uncaging pulses (summarized if Figure 4 and Figure 7). Furthermore, pooling data across all spines, we did not observe differences in spine length with respect to depth in the slice. Therefore, given precautions that we have taken as described above, we are confident that any systematic biases in the effective uncaging power are minimal under our experimental conditions.

2) Spines with long necks are described as having immature (weak) synapses and high resistance necks that would filter EPSPs. It's not possible to attribute the small EPSPs recorded at the soma to spine neck resistance without knowing if there are AMPARs in the spine head and, if there are AMPARs, how the receptor number compares to that of shorter neck spines. Experiments should be conducted to measure/estimate AMPAR and NMDAR numbers for spines of varying lengths. Fluor intensity of PSD-95-venus puncta might be a suitable proxy.

As the reviewer suggested, we measured PSD-95-venus fluorescence in spines of varying lengths. We reanalyzed the z-stacks we used to measure spine and shaft synapse densities. We observe a weak, though highly significant correlation between mVenus intensity and spine length. These new data have been added to the paper (Figure 7). Notably, a significant number of long spines displayed mVenus fluorescence that was significantly higher than the average background mVenus intensities measured in the dendrite. Therefore, using PSD-95 fluorescence intensity as a proxy for AMPAR density, our new analysis of the mVenus fluorescence shown in Figure 7 support the idea that AMPAR are indeed present on longer spines albeit at lower densities than on shorter spines.

In addition, we have performed an analogous set of voltage-clamp experiments (described in our response to Point #5) in which we recorded AMPAR-mediated uEPSCs from long and short spines in the presence of cyclothiazide to limit AMPAR desensitization. Similarly, we find that longer spines exhibit smaller amplitude uEPSCs, suggesting that AMPARs are present on long spines but at lower densities relative to short spines.

3) FRAP analysis needs to be revisited with fits constrained to baseline fluorescence. The authors use FRAP measurements (proxy for chemical compartmentalization) to estimate spine neck resistance. In the FRAP example shown, 15-20% of the fluorescence signal does not recover, yet the authors fit this trace with a monoexponential function, that is not constrained to baseline fluorescence and report a tau of 30 ms. (This trace is better described with a double exponential, with the first time constant possibly reflecting diffusion of fluorophore within the spine and the second reflecting diffusion between the spine and dendrite.). If the fluorescence of an untethered molecule doesn't recover, there is a stronger barrier to diffusion than reported. How many of the data points are affected by this?

To address this point, we will first discuss data fits to single exponentials. Without constraining the exponential fits to baseline fluorescence values, we found that the average steady-state value approached by the exponential fit (y_0_)was ~95% of the baseline fluorescence signal. A histogram of the original values is provided below. In the example shown in the original manuscript (Figure 6) the unconstrained y_0_ was 92% of the baseline signal.

After reexamining these data, it appears that some spine fluorescent signals do not make a complete return to baseline due to gradual bleaching of the Alexa-594 dye by the imaging laser. Acquisition-related bleaching and different means of correction have been described in a variety of FRAP experiments (Ishikawa-Ankerhold et al., 2012). We were able to estimate the extent of acquisition-related bleaching for each spine using scans in which we did not target the uncaging laser to the spine head. These data were routinely collected when we uncaged glutamate near the spines to measure uEPSP amplitude and Ca^2+^ influx. The extent of this acquisition-bleaching was roughly proportional to the incompleteness of recovery in FRAP experiments (on average 3.5 ± 0.7% over 33 spines). Alexa 594 signals from the “acquisition-only” scans were fit with a straight line. FRAP data were then normalized to these fits.

After performing this correction we reanalyzed our data by constraining double exponential fits to return to the baseline fluorescence as the reviewer suggested. Consistent with the reviewer’s expectation, constraining fits to baseline fluorescence generally increased the measured time constant, suggesting a higher barrier to diffusion than we previously reported. Correspondingly, the estimated spine neck resistances are larger.

We also considered the reviewer’s suggestion that the fast time constant of double-exponential fits could reflect diffusion of Alexa within the spine head. We found no correlation between spine head volume and the rate of the fast time constant (R=0.03) or the relative weight of the fast time constant to the overall fit (R=0.01). Therefore, we used a weighted time constant from the double-exponential fits to describe the diffusional coupling between the spine and dendrite. Similar to our original analysis with mono-exponential fits, we see a significant correlation between the time course of FRAP and spine length (R=0.52; p=0.002) as well as between the uEPSP amplitude and estimated spine neck resistance (R=-0.50, p=0.0046). A new example FRAP trace is provided. The trace was normalized to baseline fluorescence to aid readers in observing the extent of initial bleaching and recovery.

Additionally, spine neck resistance is proportional to the length cross sectional area axial resistance. The spine neck diameter is going to have a much larger impact on resistance than length. Finally, the relationship that is most relevant, and least subject to assumptions based on unmeasurable physical features (at least with 2P) would be the FRAP versus uEPSP amplitude. A better experiment altogether would be to uncage (in a standardized way) on either side of the neck and measure a voltage dependent process in the spine head. This has been done by the Sabatini and Magee labs to good effect.

We agree that the cross sectional area of the spine will largely determine chemical diffusion and the spine neck resistance, but is not adequately measurable with 2P imaging. As suggested by the reviewer, the FRAP vs. uEPSP amplitude is probably the most suitable for two-photon. When considering the FRAP data, however, it is important to note that the time course of FRAP will depend on both the morphology of the spine neck as well as the volume of the spine head. As described in the studies by Tonnenson et al., and Takasaki and Sabatini, τ = (head volume*neck length)/(diffusion constant*neck cross sectional area).

Based on the equations above, the FRAP time course is not only dependent on the spine neck resistance, but also on the spine head volume. Therefore, a measure that incorporates both the FRAP time course and the spine head volume (such as the estimates of R_neck_) will be a better indicator of the electrical/diffusive barrier of the spine neck than FRAP alone. Consistent with this, measuring the relationship between uEPSP amplitude and FRAP time course indicates a weak correlation that nears statistical significance (R=-0.34, p=0.065). However, there is a moderate and highly significant correlation between uEPSP amplitude and the estimated neck resistance (R=-0.50, p=0.0046). These estimates of neck resistance are not based on morphological measurements of neck diameter or neck length, but only the spine head volume.

Estimating spine head volume with 2P has been used by multiple labs to measure relative differences between spines or changes in spine volume with time, though admittedly these estimates of are not as accurate as those made using super resolution light microscopy or EM. Despite these differences, we point out here, as we do in the manuscript, that the estimates of spine neck resistance from our data are similar to those utilizing higher resolution STED imaging to measure spine head volume (Tonnenson et al., 2014).

*4) The authors report there is an increase AMPA to NMDA ratio in shorter spines. The data presented in 6C indicate that on average the ratio is ~ equivalent.*

In the text of the original manuscript, we point out that the amplitude of both AMPAR and NMDAR mediated currents in longer spines. Importantly, however, we also find that the relative decrease in the AMPAR mediate current amplitude with spine length is more dramatic, resulting in the reduction in AMPA/NMDA ratio with increasing spine length reported in Figure 6.

In the revised version of this manuscript, we provide a new pair of currents from example spines to better illustrate the summary data (Figure 6). In addition, Figure 6 has been redrawn with the AMPA and NMDA data plotted on neighboring plots to make the subtle differences in the relationship between current amplitude and spine length more clear.

Moreover, the authors need to distinguish between 1) differences in receptor number at synapses on long versus short spines, 2) differential filtering of fast AMPA versus slow NMDAR mediated potentials, and 3) over all compartmentalization of electrical and biochemical signals within the spine, via the spine neck. Specifically, are AMPA and NMDA electrical signals equally attenuated by the spine neck?

As the reviewer points out, the spine neck could perhaps more dramatically attenuate AMPA signals than NMDA signals due to differences in receptor kinetics. To address this concern, we performed additional analysis of existing data by measuring the relationship between spine length and the AMPAR-mediated charge transfer (the integral of the currents measured at -70 mV). The spine neck resistance is predicted to have a much smaller effect on total transfer of charge to the dendrite than on current amplitude. These data have been added to the manuscript (Figure 6).

Furthermore, we conducted additional experiments (Figure 6) in which we isolated AMPAR currents and blocked desensitization of the AMPARs with cyclothiazide. This treatment has the effect of increasing the total synaptic conductance and slowing the kinetics of the AMPAR ~2.5 fold. By increasing the synaptic conductance, we minimize the effect of the neck resistance on the measurements of current and charge transfer. Furthermore, by slowing the kinetics of the receptor we can mitigate the low pass filtering that the spine neck may produce. In these conditions, we observe a significant correlation between spine length and AMPA current amplitude as well as spine length and charge transfer.

Combined with our additional analysis of the PSD-95 ENABLED mouse (described in Point #2) we have significant evidence to suggest that a lower AMPAR density contributes to the small uEPSPs produced by long spines.

5) Experiments testing the ohmic nature of spine neck resistance should be shown as well as described in text.

In the original manuscript, we stated that the spine neck resistance was predicted to be ohmic as we believe it to be primarily dependent on the morphology of the spine neck and the axial resistance of the cytoplasm (as described by the reviewer in point 3). We assumed that it was unlikely that these parameters would readily change with transmembrane voltage. To test this assumption, we measured isolated AMPAR currents in cyclothiazide (described above) at -70 mV and +40 mV (NMDA receptors were blocked in these experiments). If the spine neck resistance were non-ohmic, the relationship between neck length and the AMPAR-mediated current would differ between positive and negative voltages. To look at this, we plotted the ratio of AMPAR current amplitude at -70 and +40 mV against spine length (Figure 6). We observe no correlation between this ratio and spine length, suggesting the spine neck resistance is indeed ohmic.

6) If PSD-95-venus fluorescence correlates with uEPSP amplitude (and presumably AMPAR number) how does venus intensity correlate with spine length? This may provide a strong paradigm for disambiguating the issues in points (3) and (4) The authors might have these experiments done already but only need to analyze them. If the long spines have no PSD-95 puncta, they might not have AMPARs, so stating the spine neck filters these currents is inaccurate.

See response provided in Point #2.

7) The authors uncage on the spine and dendrite while the cell is at different membrane potentials. They observe the spine Ca signal is larger at more negative membrane potentials and attribute this to increase relief of Mg block at spine synapses due to the larger depolarization of the spine head. Although this is a possible interpretation, it is also possible that there are T-type VGCCs in the spine head that are not present in the adjacent shaft. Experiments should be done to determine if the spine and dendrite have the same NMDAR and VGCC mediated Ca signals. If the Ca sources are different the measurements they've made might not be correctly interpreted.

To address the reviewer’s concern that T-type VGCCs are present in the spine head but not in the dendritic shaft, we repeated these experiments in the presence the specific T-type channel block TTA-P2 (n=16 spines and 16 shaft synapses) and obtained results similar to our original dataset (see comparison below). In the presence of TTA-P2 at voltages below -60 mV, normalized spine Ca^2+^ influx was significantly smaller than normalized shaft Ca^2+^ influx (p=0.004). Given the similarities of the data sets, they have been pooled in the resubmitted manuscript.

8) What does the Ca signal in the spine and dendrite look like during tonic firing without uncaging? The authors should show Ca transients from spikes alone.

We have performed additional experiments and analyses of these data as described in response to Reviewer 1. Figure 9 has been updated to reflect this.